# Technical note: GUARD – An automated fluid sampler preventing sample alteration by contamination, evaporation and gas exchange, suitable for remote areas and harsh conditions

Arno Hartmann[1], Marc Luetscher[2], Ralf Wachter[1], Philipp Holz[1], Elisabeth Eiche[1], Thomas Neumann[3]

[1]Institute of Applied Geosciences, Karlsruhe Institute of Technology, Karlsruhe, D-76131, Germany
[2]Swiss Institute for Speleology and Karst Studies (SISKA), La Chaux-de-Fonds, CH-2301, Switzerland
[3]Institute of Applied Geosciences, University of Berlin, Berlin, D-10587, Germany

*Correspondence to*: Arno Hartmann (arno.hartmann@kit.edu)

**Abstract.** Automated water sampling devices adapted to field operation have proven highly useful for environmental research as well as in the public and private sector, where natural or artificial waters need to be tested regularly for compliance with environmental and health regulations. Such autosamplers are already available on the market in slightly differing versions, but none of these devices are capable of sealing the collected samples to prevent sample alteration by contamination, evaporation or gas exchange. In many sampling cases, however, this feature is essential, for instance for studying the hydrological cycle based on isotopes in rainwater, or for monitoring waters contaminated with toxic gases or other volatile compounds detrimental to biota and human health. Therefore, we have developed a new mobile autosampler, which injects water samples directly into airtight vials, thus preventing any sample alteration. Further advantages include low production costs, compact dimensions and low weight allowing for easy transport, a wide range of selectable sampling intervals as well as a low power consumption, which make it suitable for long-term applications even in remote areas and harsh (outdoor) conditions due to its heavy-duty water-proof casing. In this paper, we demonstrate 1) the sampler's mechanical functioning, 2) the long-term stability of the collected samples with regard to evaporation and gas exchange and 3) the potential of our device in a wide variety of applications drawing on laboratory and field experiments in different karst caves, which represent one of the most challenging sampling environments.

## 1 Introduction

Sampling natural waters (water bodies and rainwater), as well as artificial waters (including process and waste water) for subsequent hydro- and geochemical analysis, is indispensable for understanding the behaviour of natural and artificial waters, for determining their spatial and temporal variability (e.g. in the concentration of toxic compounds) as well as for determining their current state (e.g. Hiscock and Bense, 2014). A thorough monitoring of water is thus important, not only for scientific purposes but also to prevent adverse effects on human health and ecosystems, whether in the short, medium or long term.

However, the manual collection of water samples is not only time-consuming but also expensive and logistically challenging. This is particularly true in remote areas with poor or no infrastructure, where expenses for field trips and equipment transport to the sampling site and back are often significantly increased. Furthermore, in most cases the properties and composition of the water under investigation change with time, for instance, the concentration of pollutants like heavy metals or polycyclic aromatics (Appelo and Postma, 2005). In these cases, it is necessary to repeat water sampling multiple times at short intervals and over a sufficiently long time period, increasing the logistical constraints (e.g. Ahuja, 2013). When it comes to long-term monitoring of a site at relatively short intervals, manual sampling is definitely no viable option anymore (Chapin, 2015).

This impasse can only be overcome by automation: Autosamplers suitable for field operation, such as the portable sampler 3700C Compact (Teledyne ISCO, USA) are already available on the market and offer the opportunity to automatically and repeatedly sample water bodies (oceans, estuaries, lakes, rivers, groundwater, etc.). As they can be powered by batteries or solar panels these autosamplers do not need a connection to the grid and can, therefore, be applied even in remote areas.

Despite their suitability for a wide range of applications, available autosamplers lack the capacity to automatically seal the sample vials after collection. This, however, is absolutely essential in many sampling applications where any exchange of gases and/or volatile components between the sample and the ambient air needs to be prevented in order to preserve the sample's original properties until analysis. Examples are numerous and include cases of water being contaminated by toxic gases originating from anaerobic degradation of organics (e.g. $CH_4$ and $H_2S$) or by volatile organic compounds (VOCs), such as tetrachloroethene, benzene, MTBE or formaldehyde (e.g. Reemtsma and Jekel, 2010).

Sample vials also need to be gastight directly after sample collection where evaporation or condensation has to be impeded to prevent sample alteration, for example where non-volatile components like cations and anions need to be quantified. In such cases evaporation would cause erroneously augmented concentration values (Hiscock and Bense, 2014), especially in long-term monitoring schemes.

Similarly, the prevention of evaporation is paramount in all studies investigating the hydrological cycle based on the stable isotopes of hydrogen and oxygen (indicated as $\delta D$ and $\delta^{18}O$ values). The majority of such studies rely on rainwater samples (mostly) collected manually at stations of the Global Network of Isotopes in Precipitation (GNIP; IAEA/WMO, 1994) coordinated by the International Atomic Energy Agency (IAEA) with the sampling performed by dedicated partner institutions in member states of the IAEA or the World Meteorological Organisation (WMO). At these stations, rainwater is generally sampled at monthly resolution to ensure worldwide compatibility of GNIP data from different sources. While most of these samples are collected manually, a number of active or passive totalizers compliant with the GNIP sampling guidelines (Terzer et al., 2016) are in operation at GNIP stations without permanent staffing. Manual sampling at higher temporal resolution, such as rainfall event-based sampling, is practically impossible as this would require round-the-clock stand-by duty.

Furthermore, sample alteration due to evaporation is commonly prevented by sealing the water samples' surface with paraffin oil despite it causing an increased need for maintenance of the standard instrument for water isotope analysis, i.e. Cavity Ring-Down Spectroscopy (CRDS).

Establishing an isotope baseline for meteoric waters is crucial for research in hydrology, meteorology and other scientific fields. While remarkable progress has been made thanks to GNIP data, the network is still spatially and temporally discontinuous, among other reasons due to the practical constraints on rainwater sampling in remote areas. Automated rainwater sampling could help solve this issue and increased maintenance of spectrometers could be avoided by applying gastight sample vials. This clearly illustrates the need for automated rainwater sampling with gastight sample vials.

Furthermore, the need for automated liquid sampling in general is demonstrated by a number of technical developments by multiple groups with the aim of creating automated liquid samplers capable of sealing the samples after collection. For instance, researchers at Oregon State University have developed the "OPEnSampler" (Nelke, Selker and Udell, 2017; http://www.open-sensing.org/opensampler/) that comprises an array of 24 solenoid valves, allowing the 24 sampling containers to be sealed from the environment after sample collection. Lukas Neuhaus has developed the "Lisa Liquidsampler" (not published) that fills 48 sample vials sealed by septa (engineered membranes that permit the transfer of fluids without air contact, usually using a double-canula) using a vacuum pump via 48 separate transfer tubes. Applying a new automated precipitation collector obtaining 96 sequential 15-mL samples, Coplen et al. (2008) were able to measure a strong decrease of 51% in the hydrogen isotope ratio ($\delta D$) of precipitation over only one hour resulting from the landfall of an extratropical cyclone along the coast of California. Evaporation and subsequent isotopic fractionation was minimised by a Teflon-coated vial cover, thus sample vials are not sealed individually.

In addition to these newly developed liquid autosamplers that are 1) suited for field operation in remote areas and under harsh (outdoor) conditions and 2) capable of sealing the sample vials (gastight) directly after sample collection, we have designed, constructed and tested a new autosampler ("GUARD") that also fulfils these requirements, but can be equipped with up to 160 sample vials due to its space-efficient design. In addition to fulfilling the former requirements, the GUARD autosampler offers further advantages including a low weight, low cost (~ 1,000 €), compact dimensions, easy transport, a wide range of selectable sampling intervals as well as a low power consumption, which permits either high-frequency sampling (e.g. every minute), long-term monitoring (e.g. 6 months), or medium-term monitoring at medium sampling frequency (e.g. daily sampling for 48 days). Therefore, the GUARD autosampler is applicable for a wide spectrum of sampling purposes, ranging from studies on characterization of high-frequency variabilities to long-term changes in natural or artificial waters. By sealing the sample vials with septa (engineered membranes that permit the transfer of fluids without air contact, usually using a double-canula), not only undesired gas exchange and evaporation/condensation are prevented, but also any contamination that might otherwise occur, especially in long-term monitoring projects where sample vials may need to stay at the sampling site for several months or longer. This protection is further ensured by a water-tight and airtight sampler case, which prevents damage from extreme weather conditions (e.g. water or dust ingress, high humidity, etc.) and also protects the samples from any external interference, e.g. from animal activity.

The setup and design of the GUARD autosampler are described in Sect. 2. In Section 3 we draw on test runs in the "laboratory" and in a karst cave, one of the most challenging sampling environments, to demonstrate 1) the mechanical functioning and 2) the long-term chemical stability of the collected samples. In Section 4 we demonstrate the potential of our autosampler in a wide variety of applications by presenting the results of a 5-day case study at a sampling interval of only

four hours, which would have been hardly possible without the use of our device. In Section 5 we compare our invention to already existing autosamplers suitable for field operation and conclude with highlighting potential sampling applications.

## 2 Methodology: Autosampler set-up and design

### 2.1 Hardware design and sampling process

The main components of the GUARD autosampler comprise an intake hose, a peristaltic pump, a mobile injection system

and a vial holder (Fig. 1, Fig. 2, Tab. 1, Supplementary S5). To prevent any sample alteration resulting from contamination, evaporation, condensation and/or gas exchange during sample storage, the fluid samples (12 mL) are injected into air/gastight vials using a peristaltic pump, at a user-defined date, time and interval.

At the beginning of the first sampling interval, 12 mL of fluid are sucked into the autosampler's tubing made of flexible and chemical-resistant FKM tubing that is hydraulically connected to the water under investigation (e.g. a lake). As the tubing is

just long enough to accommodate precisely 12 mL (i.e. 1247 mm), the sample is not yet injected into the corresponding vial, but at first remains inside the tubing where it is already protected from gas exchange.

At the beginning of the subsequent sampling interval, two electric motors move the sampler's X- and Y-slide via toothed rubber belts until two separate end-switches are triggered that provide positioning calibration. Both slides are then positioned directly above the first sample vial. After a 2-seconds safety delay, a servo screwed to the sampler's Z-slide moves the Z-

slide down, until a metal double cannula attached to the front end of the FKM tubing just barely pierces the rubber septum which keeps the vial permanently air/gastight. After another 2-seconds safety delay, the sampler's pump is reactivated and the collected water is injected into the vial through one of the cannulas (the "sample cannula") while the subsequent sample is sucked into the sampler's tubing simultaneously. As a result of this design, the sample injection always lags the sample collection by one interval.

As the sample vials are airtight, an overpressure builds up inside the vials during sample injection. Pressure equalisation is achieved via the second of the two cannulas which are soldered to one another. To achieve the maximum sample volume (here: 12 mL) this "pressure release cannula" is located 2-3 mm above the sample cannula, and the vials are filled with an overflow of several droplets. This setup avoids any unwanted interaction (e.g. gas or isotope exchange) between the collected fluid sample and the supernatant air/gas left inside the vial. Subsequent to a 10-seconds safety delay implemented

to allow for complete pressure equalisation and sample injection, the Z-slide with the double cannula is moved back up to its home position and the X- and Y-slides are positioned above the next sample vial. After another 2-seconds safety delay, the sampler enters a hibernation mode to minimise power consumption until the hibernation is interrupted with the start of the

next sampling interval. After completion of a full sampling sequence, the Z-slide moves back up to its home position, the X-slide moves to its end position and the sampler waits for input from the operator.

In the setup shown here the collected samples have a volume of 12 mL which is sufficient for most analyses, including isotope ratio mass spectrometry (IR-MS) and inductively coupled plasma mass spectrometry (ICP-MS). The pumping process takes only about 22 seconds and, thus, the collected sample represents the water under investigation at a given instant (integrated over 22 seconds). As one entire sampling step takes only 41 seconds (power consumption: 2.1 mAh), the autosampler is capable of high-resolution fluid sampling with a minimum interval of one minute. This is valuable where high-frequency variations in the composition of the sampled fluid need to be resolved, for instance, in artificial tracer tests at the onset of the tracer breakthrough where samples are commonly collected at intervals as short as one minute (Leibundgut et al., 2009). If needed, the sample volume can be modified by changing the duration of the pumping step. For example, to obtain a 100 mL sample the pumping step would take about 3 minutes.

Using a 12 V battery with a capacity of 40 Ah the GUARD autosampler can operate off-grid for about 100 days without interruption at a 2-day interval (one full sampling sequence), thanks to the hibernation mode during which power consumption is reduced to 16.5 mA (we are currently working on further reducing the power consumption). On such long time spans the power consumed during the actual sampling process is practically negligible. If longer operation durations are necessary, multiple batteries can be connected in parallel to increase the total capacity. The sampler can also run on 12 V Li-ion batteries if weight is an important constraint. Additionally, nearly discharged batteries can be replaced with fully charged ones without interrupting a running sampling sequence by using an electrical bypass. Of course, implementing an appropriate rectifier, the autosampler can also run on mains power in which case runtime limitations do no longer apply.

Fig. 2 shows the GUARD autosampler in detail and in operation during a 5-day case study (Sect. 4) focusing on the carbon isotope geochemistry ($\delta^{13}C_{DIC}$) of dripwater originating from a stalactite in a karst cave in northern Bavaria, Germany.

## 2.2 Electronic and software design

Most of the autosampler's electronic components are accommodated in the control unit (Fig. 1) inside an additional casing for protection. The centrepiece of the electronic design is the Arduino® Mega 2560 board which is based on an Atmel ATmega 2560 microcontroller. This microcontroller enables the autosampler to enter the hibernation mode during which power consumption is reduced 50-fold as compared to the power consumption during slide movement. It also contains a non-volatile 4 KB EEPROM memory in which the data (time and position) of the previous injection are saved temporarily. The interrupts of the hibernation mode at the beginning of each sampling interval are triggered by a real-time clock (RTC) chip that includes a separate 3V lithium button cell battery which ensures that the program controlling the sampler operation remains active, even if the main power supply may be interrupted. The electrical circuit diagram as well as a Bill of Materials are given in the Supplementary (S1, S6).

The program controlling the autosampler was constructed with the open-source software Arduino (version 1.8.3). The code is written in Java and can be uploaded to the board via a USB connection. A flowchart illustrating the operation of the GUARD autosampler is shown in Fig. 3.

## 3 Demonstration of the autosampler's functioning

### 3.1 X-Y-positioning

During and after the development of the GUARD autosampler we have conducted various indoor experiments to test the mechanical functioning of our prototype. One important requirement in that respect is a precise and reliable positioning of the X- and Y-slides at the exact locations of the sample vials. High precision is especially important for an efficient use of the space available for the sample vials which are arranged as close as possible to each other (yielding a capacity of 160 vials at the given casing dimensions). To achieve this high precision, all movements are executed by Computerised Numerical Control (CNC). The stepper motors for the slide movement in X- and Y-direction are programmed to turn in quarter-steps which correspond to a rotation of only 0.45 °. Consequently, the slide movements along the X- and Y-axis are accurate to less than 1 mm. As the pierceable area of the rubber septa is 7 mm in diameter, there is a more than sufficient error margin of about 350 %. This prevents the double cannula from hitting the sample rack during the Z-slide's downward movement which would cause the double cannula to deform and the respective sampling sequence to fail.

### 3.2 Sample injection

Another important aspect of an error-free mechanical functioning is a successful sample injection with an optimal use of the available sample volume. To demonstrate the fulfilment of this prerequisite we ran a complete sampling cycle comprising 48 tap water "samples" and analysed the size of the air bubbles remaining in the vials after sample injection. The results of this test are shown in Fig. 4: Most bubbles are 9.5 mm in diameter or less which confirms that the vials (internal diameter $\leq 15$ mm) are filled quasi-completely during sample injection. Assuming the bubbles were a perfect sphere, they would make up $\leq$ 0.45 mL (or $\leq 4$ % of the inner vial volume). Considering that the bubbles are in fact strongly flattened, they more like make up $\leq$ 0.22 mL (or $\leq 2$ % of the inner vial volume). Therefore, any sample alteration due to interactions between the fluid sample and the supernatant air/gas, for instance, due to isotopic exchange, is prevented.

### 3.3 First field test: Comparison of automatic and manual samples & long-term sample stability

With respect to the quality of the samples collected by the GUARD autosampler, there are two main prerequisites: 1) Samples need to yield identical analytical results, whether they are collected automatically or manually, and 2) samples need to be unaltered and stable, even in the long term, which is ensured by the sample vials being air/gastight. In order to ascertain the fulfilment of both of these criteria, we applied our device in a first field experiment in a karst cave to automatically sample the water at a specific drip site over the course of 33 days at daily intervals. For comparison, we collected 12

dripwater samples manually. We then analysed the oxygen isotopes in these water samples with cavity ring-down spectroscopy (CRDS) on a liquid water isotope analyser (LWIA-24d; Los Gatos Research). The standards used for calibration were LGR1A, USGS 46 and USGS 48. The accuracy (<0.07 ‰) was tested by repeated measurements of the control standard material LGR 2C. The average precision of the individual measurements (n = 124) was ±0.4 ‰. The isotope data are shown in Fig. 5 and Fig. 6.

Fig. 5 demonstrates that the oxygen isotope results from the automatically and the manually collected samples are in good agreement with each other, especially considering the respective analytical error ranges. In two cases (13[th] of December, around 3 pm and 22[nd] of December 2016, around 4 pm), $\delta^{18}O$ values do not seem to agree within error at first. However, as the manual samples had to be collected at least 15 minutes before or after the automatic collection in order to allow for sufficient sample volumes (due to the low drip rate at this specific site), this seeming mismatch can be explained with the high-frequency variability of the dripwater $\delta^{18}O$ values at this drip site. This is best exemplified with the last two automatically collected samples in the left sub-plot in Fig. 5, where $\delta^{18}O$ values dropped from -10.05 ‰ to -10.24 ‰ within only 30 minutes. The sample collected manually exactly in between these two yields an intermediate $\delta^{18}O$ value of -10.06 ‰ and is therefore consistent with the automatically collected samples. More importantly, there is no systematic discrepancy between the automatically and the manually collected samples, with the respective arithmetic mean $\delta^{18}O$ values, calculated for the entire sampling period (both sub-plots of Fig. 5), differing by only 0.03 ‰. The results for δD are similar to the $\delta^{18}O$ results and also confirm the long-term stability of the samples (Supplementary S3).

In order to demonstrate that the sample vials are completely airtight and remain so even after the double cannula has pierced the rubber septa during sample injection, we measured the oxygen isotopic composition of nine different samples (stored in a fridge at 11.2 °C) repeatedly over a time interval of six months. The results (Fig. 6) confirm the long-term stability of the samples: If the vials were not airtight, evaporation would have led to a preferential removal of isotopically light water molecules from the water samples due to their higher vapour pressure (e.g. Hoefs, 2015) and, consequently, to an increase of the $\delta^{18}O$ value of the remaining water sample over time. Such a positive trend is not present in the data and the results from the repeated measurements agree well with the initial ones. The difference in $\delta^{18}O$ values between initial and repeated measurements ranges from 0.00 ‰ (lt02-05) to 0.15 ‰ (between 2[nd] and 3[rd] measurement of sample lt03), but averages out at -0.01 ‰ over all measurements (median also -0.01 ‰) indicating that there is no systemic discrepancy between initial and repeated analyses. The results for δD are similar to the $\delta^{18}O$ results and also confirm the long-term stability of the samples (Supplementary S2).

To provide the reader with a notion of the effect of evaporation on the sample $\delta^{18}O$ values, we have calculated both evaporation and $\delta^{18}O$ change for the conditions prevalent in our fridge (temperature 11.2 °C, relative humidity 24 %) and assuming an opening of the sample vial of 5 % to imitate a minor lack of airtightness. The results of these calculations (Supplementary S4) demonstrate that even a small slit in a sample vial's rubber septum equalling only 5 % of the vial's inner cross section leads to a substantial shift towards higher $\delta^{18}O$ values in the residual water over time. After three months (90 days), for instance, $\delta^{18}O$ values would have risen from -10.1 ‰ by about 1.3 ‰ to -8.8 ‰. For comparison, the difference

between the lowest and the highest $\delta^{18}O$ value in Fig. 6 is still below 0.3 ‰, while those data points span an even longer period of six months. Most importantly, there is no positive trend in the $\delta^{18}O$ values in Fig. 6 which illustrates that the sample vials are sealed properly, even after sample injection.

## 4 Case study: High-resolution drip sampling for speleothem science

The potential and usefulness of our autosampler are demonstrated in a first case study that would have been both too expensive and time-consuming to conduct without our device. The goal was 1) to prove the existence of high-frequency (daily) variability in the carbon isotope values ($\delta^{13}C$) of dissolved inorganic carbon (DIC) in cave dripwaters and 2) to quantify its amplitude. This variability has important implications for the reconstruction of past environmental changes from speleothem $\delta^{13}C$ values as these are not only a function of the dripwater $\delta^{13}C$ signal originating from the surface

environment, but also of the intensity of degassing of excess $CO_2$ from the dripwater (Fairchild et al., 2006). However, to date $\delta^{13}C_{DIC}$ variability has only been documented on the seasonal and annual scale (e.g. Spötl et al., 2005; Mattey et al., 2010), certainly also due to the lack of practical solutions for the high-frequency dripwater sampling in caves.

The existence of such high-frequency variability in $\delta^{13}C_{DIC}$ can be postulated based on the knowledge that cave air $CO_2$ concentrations can vary both strongly and quickly (e.g. Luetscher and Ziegler, 2012) as a response to ventilation processes

(e.g. Tremaine et al., 2011): In general, strong ventilation of a cave system leads to an input of low $CO_2$ ambient air which (partly) replaces the $CO_2$ enriched cave air. The lowered $CO_2$ concentration causes enhanced degassing of excess $CO_2$ from the cave dripwater, which, in turn, results in increased dripwater $\delta^{13}C_{DIC}$ values, as isotopically light $CO_2$ transits preferentially from the liquid to the gas phase (Clark and Fritz, 1999).

### 4.1 Study area

This case study was carried out in the "Kleine Teufelshöhle" (N 49°45′17″, E 11°25′12″) in the Franconian Switzerland region in northern Bavaria, Germany. This cave is characterized by dynamic ventilation (forced convection). The mean annual air temperature is around 8°C and the air humidity is close to saturation. This site warrants conditions suitable for a demanding field test due to 1) the lack of an electric supply network and to 2) the high relative humidity which poses challenges for electrical appliances in general. The GUARD autosampler was placed at a location adequate for sampling the

dripwater from a specific group of stalactites, at drip site "DS4" (Fig. 2, right).

### 4.2 Materials and methods

Dripwater sampling was conducted automatically at 4-hour intervals over a period of five days yielding a total of 22 samples. The stable carbon isotopic composition of the dripwater DIC was determined at the University of Innsbruck using continuous-flow isotope ratio mass spectrometry following the method described in Spötl (2005). Calibration of the raw

results versus the V-PDB scale is achieved using in-house calcite standards (subsequent to linearity correction) that have

been calibrated against NBS-18, NBS-19, CO-1 and CO-8 reference materials. The external precision calculated over 12 standards per run is typically ≤0.07 ‰ for $\delta^{13}C$.

Cave air $CO_2$ concentrations were logged every 30 minutes with a Vaisala GM70 hand-held unit equipped with a $CO_2$ probe optimised for the 0–2000 ppmV range (GMP222; accuracy: ± 1.5 % of the calibration value plus 2 % of the measured value).

Cave air temperature (T) and relative humidity (RH) were logged at 10-minute intervals with a Tinytag TGP-4500 (Gemini Loggers; accuracy: ± 0.5 °C at 8 °C and ± 3.0 % RH at 25 °C), while the combined drip rate of the stalactite cluster was logged with a Stalagmate Mark 3 (TGC-0011; Driptych) and integrated over 5-minute increments.

### 4.3 Results

Over the duration of the 5-day case study, RH was constant at 100 % and the drip rate oscillated between 26 and 28 drops

per 5-min increment (5.2 to 5.6 drops/min). The results of the T, $CO_2$ and $\delta^{13}C_{DIC}$ analyses are summarised in Fig. 7. Cave air $CO_2$ concentrations range from 520 to 1430 ppmV, with an average of 748 ppmV (n = 476) and a median of 710 ppmV, and $\delta^{13}C_{DIC}$ values range from -9.8 to -7.7 ‰, with an average of -8.8 ‰ (n = 22) and a median of -8.9 ‰. $CO_2$ concentrations peaked thrice (circled numbers in Fig. 7), with two smaller peaks of 910 and 1030 ppmV at the beginning of the monitoring period being followed by the most prominent and broad peak of 1430 ppmV that occurred during the night

from the 2$^{nd}$ to the 3$^{rd}$ of May 2017. All three $CO_2$ peaks, particularly the last one, precisely coincide with troughs in the $\delta^{13}C_{DIC}$ values, while $CO_2$ troughs coincide with $\delta^{13}C_{DIC}$ peaks, which results in a distinct negative correlation (Spearman's $\rho$ = -0.88) of both geochemical signals. Temperature varies only very slightly, with both average and median being 8.6°C (n = 1425). Despite the low amplitude of T variations, T appears to correlate positively with $\delta^{13}C_{DIC}$, but only weakly (Spearman's $\rho = 0.36$).

### 4.4 Interpretations

The dripwater analyses obtained from the Kleine Teufelshöhle at 4-hour resolution over five days clearly prove the presence of a high-frequency variability in the $\delta^{13}C_{DIC}$, in addition to the already documented seasonal and interannual variability (Spötl et al., 2005; Mattey et al., 2010). In this case, the maximum amplitude is 2.1 ‰ – a change that is great enough to be resolved by state-of-the-art isotope-ratio mass spectrometers (IR-MS). While this 2.1 ‰ change occurred over a period of

almost two days (2017-05-02 01:00 to 2017-05-03 17:00), additional variability is observed on even smaller time-scales. For example, the difference in $\delta^{13}C_{DIC}$ values between the first local minimum (-9.04 ‰) and the first local maximum (-8.32 ‰) came about in only 8 hours, with an amplitude of 0.72 ‰, suggesting rapid responses to even small changes in the ventilation regime.

The strong negative correlation between $\delta^{13}C_{DIC}$ values and $CO_2$ concentrations is consistent with ventilation events that lead

to decreased cave air $CO_2$ concentrations: As high $CO_2$ cave air is partly replaced by low $CO_2$ ambient air degassing of excess $CO_2$ from the drip water is enhanced. As the process of degassing favours isotopically light $CO_2$ molecules, the $\delta^{13}C_{DIC}$ values in the dripwater are increased during these ventilation events. This interpretation also seems to be confirmed

by the measured T variations: Although they are very small, the positive albeit moderate correlation with $\delta^{13}C_{DIC}$ suggests that, during ventilation, part of the cave air is replaced by relatively warm low $CO_2$ ambient air. During winter months, it would be replaced by relatively cold but still low $CO_2$ ambient air.

In order to characterize the changes in the dripwater $\delta^{13}C_{DIC}$ with respect to the cave air $CO_2$ concentration, we have determined the amplitude of the maxima/minima in $\delta^{13}C_{DIC}$ and $CO_2$ relative to their respective overall mean, simply by subtracting each maximum/minimum from the mean. This yields a total of six maximum/minimum pairs that plot very well along a linear regression line (Fig. 8). According to this regression, a change in cave air $CO_2$ concentration of about 435 ppmV produces a change in dripwater $\delta^{13}C_{DIC}$ of 1 ‰ that is then transferred to the $\delta^{13}C$ signal in the speleothem fed by this dripwater.

We note that potential drift effects or sample alterations that might be caused by the automatic sampling process have not yet been examined in detail. Corresponding tests using check-standards of known $\delta^{13}C_{DIC}$ values will be performed in future studies.

## 5 Summary and potential applications

With "GUARD" we have developed and tested an automated water sampler suited to field operation in remote areas and harsh conditions that injects the samples into airtight vials in order to prevent sample alteration from contamination, evaporation/condensation and/or gas exchange.

In this paper, we have demonstrated its mechanical functioning, the long-term chemical stability of the collected samples and the potential of our autosampler in a wide variety of applications by presenting the results of a 5-day case study which would have been hardly possible without the use of our device. In this case study we have proven for the first time that cave dripwater geochemistry varies on much smaller time-scales than has previously been established. Applying the GUARD autosampler we have collected enough dripwater samples at high temporal resolution to make a first contribution to quantifying the effects of ventilation events on dripwater $\delta^{13}C_{DIC}$, which will help in using speleothem $\delta^{13}C$ as a proxy for palaeoenvironmental change.

To conclude we compare our prototype with the portable autosampler 3700C Compact (Teledyne ISCO, USA) which is, to our best knowledge, the only type of device similar to our prototype already available on the market. While there are also other (bigger and heavier) models from Teledyne ISCO and similar devices offered by other companies, the 3700C Compact autosampler is representative of the technical state of the art.

The most relevant properties of both autosamplers are compared in Table 2. As might be expected when comparing a prototype with a market-ready product it is evident that the GUARD autosampler lacks specific features that enhance the end-user comfort, such as rinse cycles between samples, an automatic compensation for changes in hydraulic head and different modes of sampling pacing. Both autosamplers are similar in weight and size. Almost a third of the GUARD's weight is due to the Pb-acid battery used in the presented setup. The battery can be transported separately from the

autosampler or can be replaced with lighter Li-Ion batteries to reduce weight. Most importantly, however, the GUARD autosampler is capable of collecting gastight samples in a considerably higher maximum number of sample vials. Thanks to a wide range of selectable sample frequencies and its capacity to operate for extended periods of time, the GUARD autosampler is well-suited for long-term sampling projects where a large number of samples need to be collected.

As existing autosamplers such as the 3700C Compact do feature neither gastight samples nor high numbers of sample vials, the GUARD autosampler is less competing with existing autosamplers as much as it closes a market gap in long-term monitoring where samples definitely need to be gastight with respect to air/gas exchange to prevent sample alteration. Within this sector, the GUARD autosampler offers many opportunities in various applications:

One example of such an application is the investigation of the hydrological cycle based on isotopes in all sorts of water

bodies, including rainwater. As mentioned in the Introduction, for this purpose the IAEA supplies researchers with isotope data generated from monthly composite samples of rainwater collected at the ~ 1,000 GNIP stations worldwide. If these stations were supplemented with GUARD autosamplers, much shorter sampling intervals would become possible which would enable researchers to investigate shorter-term variability in precipitation isotope systematics to improve our understanding of the underlying processes. To achieve this, sampling frequency needs to be at least high enough to resolve

different precipitation events ("event-based" sampling). For instance, only by using such event-based data Celle-Jeanton et al. (2001) were able to demonstrate characteristic differences in the isotopic composition of rainwater in the Mediterranean coastal region of France the authors attributed to different types of synoptic weather systems. As the synoptic weather situation can change rather quickly, monthly rainwater isotope data would have most likely been of insufficient temporal resolution to identify this relationship between isotope composition and synoptics. Naturally, the increased number of

samples generated by high-frequency sampling needs to be considered.

In addition, paraffin oil would not be required to prevent evaporation and increased maintenance of CRDS instruments could be avoided. The GUARD autosampler could also be applied at the ~ 750 stations of the Global Network for Isotopes in Precipitation (GNIR), also coordinated by the IAEA. Especially in very remote areas, the application of GUARD samplers would be a cost-effective solution to supplement GNIP and/or GNIR stations and it might even facilitate the installation of

new stations too remote for regular manual sample collection.

The GUARD autosampler could also be applied at the ~ 750 stations of the IAEA's Global Network for Isotopes in Precipitation (GNIR). Especially in very remote areas, the application of GUARD samplers would be a cost-effective solution to supplement GNIP and/or GNIR stations and it might even facilitate the installation of new stations too remote for regular manual sample collection.

Due to the temporally discontinuous nature of rainfall automatic rainwater sampling requires 1) sample pre-collection for temporary storage of rainwater until a sufficient sample volume is available while minimising or even preventing evaporation and 2) a detector such as a photo sensor to end hibernation and trigger sample collection once a sufficient sample volume has been provided by rainfall. For the case studies in karst caves presented in this paper we applied a specifically designed pre-collection container ("pre-collector") with an internal volume of exactly 12 mL (Supplementary

S7). During dripwater pre-collection a 3D-printed floating body inside the pre-collector would rise until it seals the pre-collector once it is completely filled with dripwater. Any dripwater in excess of 12 mL spills over through a small hole at the top of the pre-collector. It is important to note that, at its current setup, the GUARD autosampler does not comprise a sample volume detector and is therefore not suited for rainwater sampling. As automatic rainwater sampling would be beneficial in numerous applications, such a detector certainly represents a useful future extension to the current GUARD system.

Another example for a promising application are dripwater monitoring schemes in karst caves that are necessary especially, if not exclusively, in speleothem science (Ford and Williams, 2007; Fairchild and Baker, 2012). The case study presented here illustrates well the GUARD's potential for offering new research opportunities. As most cave sites are located far from researchers' offices and are often difficult to get to, there is a great need for automation in dripwater monitoring studies. The sampler's application in high-frequency (short-interval) dripwater sampling will enable researchers to identify, resolve and quantify short-term variability in dripwater geochemistry and to better understand these complex cave systems – a prerequisite for reliable reconstructions of past climates and environments from speleothem proxies.

## 6 Data availability

All data used in this study are available upon request from the corresponding author. In accordance with the principle of open science, further crucial data and information that enable the construction of a GUARD autosampler are available on the KITopen repository, including the Arduino sketch in its latest version (DOI: 10.5445/IR/1000085057), assembly instructions (DOI: 10.5445/IR/1000085058) and 3D-models of crucial components of the sampler (10.5445/IR/1000085059), such as the servo connector, the double-cannula holder and the sample rack.

*Author contributions*. Arno Hartmann, Marc Luetscher and Ralf Wachter conceptualised the device and the case studies. Ralf Wachter and Arno Hartmann built the device and tested it with the help of Philipp Holz. Marc Luetscher performed the stable carbon isotope analysis. Arno Hartmann and Philipp Holz conducted the cave monitoring (Kleine Teufelshöhle). Arno Hartmann prepared the manuscript with contributions from all co-authors.

*Competing interests*. The authors declare that they have no conflict of interest.

*Acknowledgements*. The technical developments leading to the prototype presented here were partly conducted in the framework of the Karst Water Technologies (KaWaTech) Project in Vietnam (CLIENTFE2-069) which was funded by the German Federal Ministry of Education and Research (BMBF). We would like to thank the FMER for the financial support. Furthermore, we would like to express our thanks to the cave associations "Höhlen- und Heimatverein Laichingen e.V." and "Forschungsgruppe Höhle und Karst Franken e.V." for their support and for granting us access to the cave sites. We direct

special thanks to Rolf Riek and Dieter and Annette Preu for their help. ML acknowledges the Tiroler Wissenschaftsfonds for partial funding of an early prototype (AP715009).

We would like to thank Dr. Rolf Hut and an anonymous reviewer for their comments that helped improve the manuscript. We also thank Mr. Nils Michelsen, Dr. Stefan Terzer-Wassmuth and Dr. Luis Araguás-Araguás for their helpful short
comments.

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

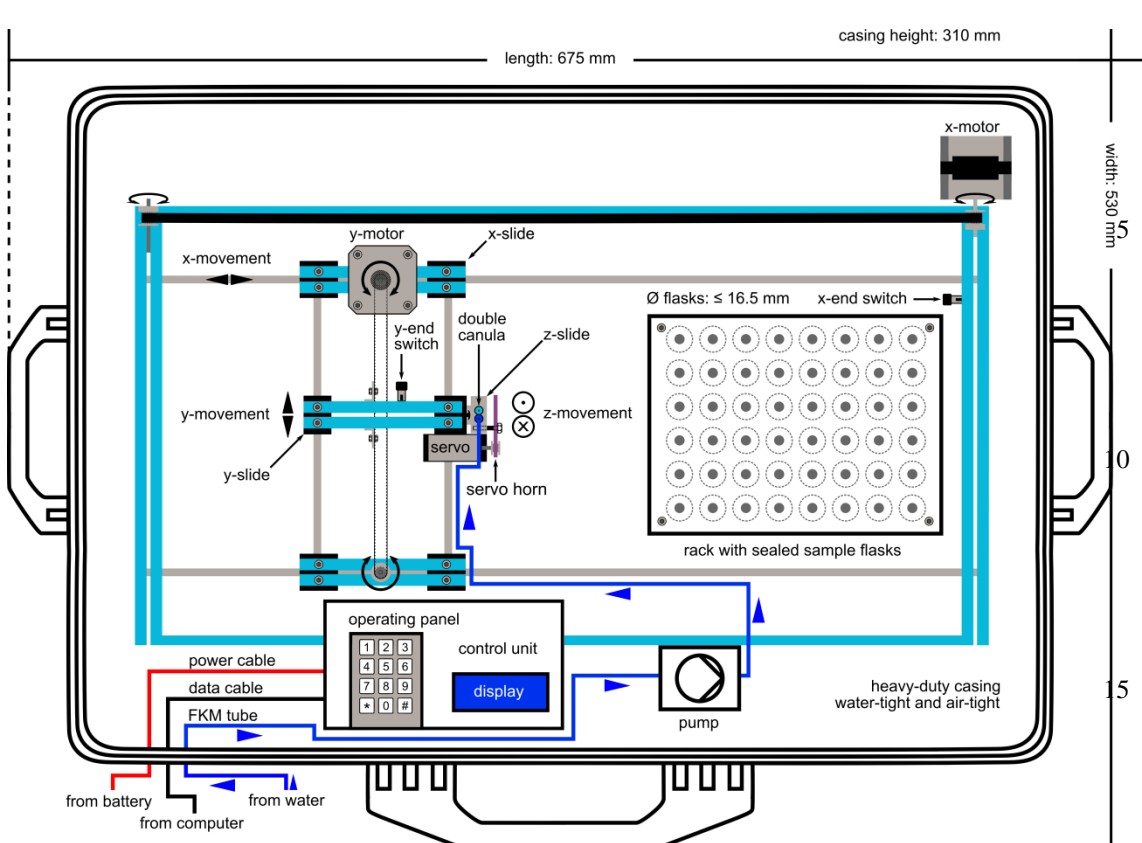

**Fig. 1.**Hardware design of the GUARD autosampler (top view, schematical). Water samples are pumped directly into vials that are permanently gastight by rubber septa. The shown set-up comprises 48 sample vials but the autosampler can be equipped with up to 160 sample vials at the given casing dimensions.

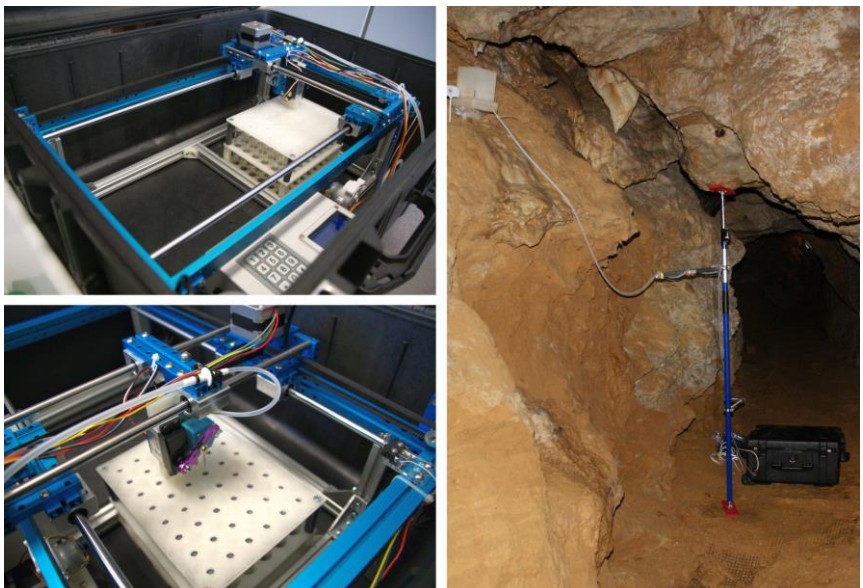

**Fig. 2. The automated fluid sampler GUARD in detail (left) and in operation (right) during a 5-day case study (Sect. 4carried out in the cave "Kleine Teufelshöhle" in the Franconian Switzerland region, Germany. At a 4-hour interval, a total of 22 dripwater samples were automatically collected for subsequent analysis of the carbon isotope values ($\delta^{13}C_{DIC}$) of the dissolved inorganic carbon (DIC).**

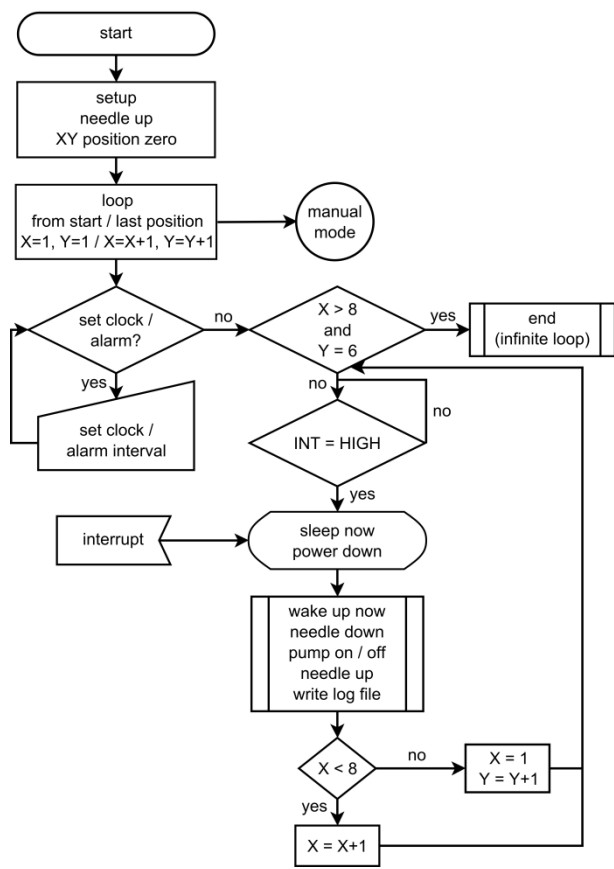

**Fig. 3.** Flowchart illustrating the autosampler's operation for a setup comprising 48 sample vials in total, arranged in lines (X direction) of 8 and columns (Y direction) of 6 vials, respectively. Once the last sample has been collected, the program enters an infinite loop and waits for input from the operator.

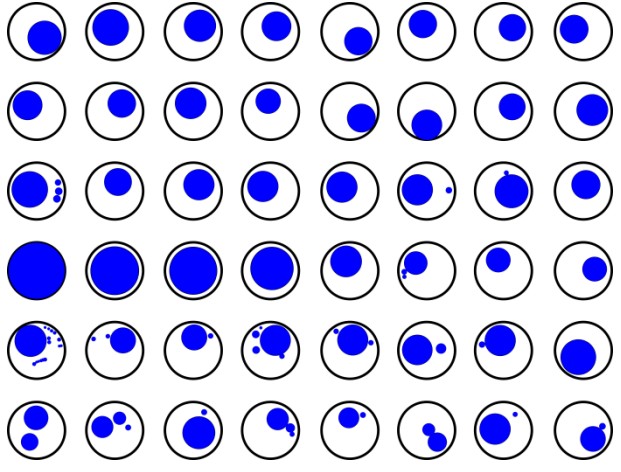

**Fig. 4.** Top view (digitised facsimile) of the sample vials (black circles) turned upside down in order to illustrate the air/gas bubbles (blue circles) remaining in the vials after sample injection.

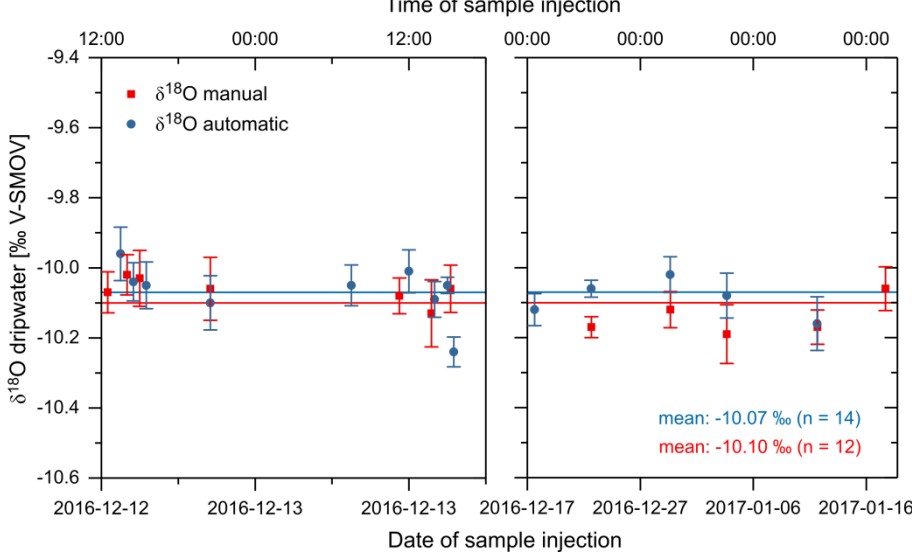

**Fig. 5. First field testing of the GUARD autosampler: Oxygen isotope values (indicated as δ<sup>18</sup>O relative to the international standard V-SMOW) in dripwater samples from a specific drip site in the karst cave „Laichinger Tiefenhöhle" in the Swabian Alb region, southern Germany. Samples were collected automatically (blue circles) over the course of 33 days (December 13, 2016, to January 14, 2017) and supplemented by 12 samples collected manually (red squares) for comparison of both methods. Error bars represent measurement uncertainty. Blue and red horizontal lines indicate the overall arithmetic mean of each data set. Note the difference in scale of the x-axes of the two sub-plots. Not all of the 33 samples were analysed for isotopic composition.**

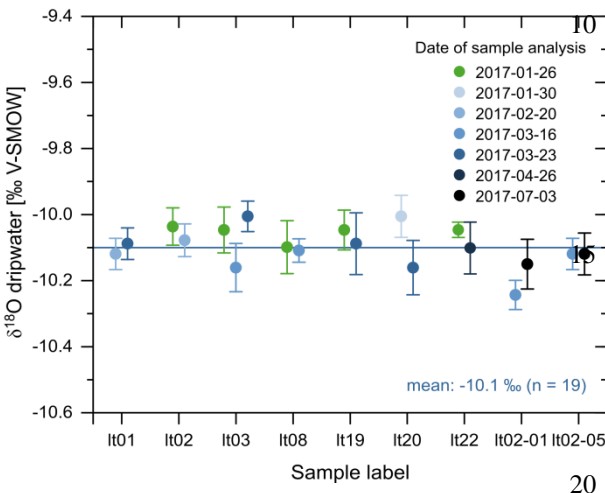

**Fig. 6. Results of repeated δ<sup>18</sup>O measurements (circles in tones of blue) measured in the automatically collected samples together with the original δ<sup>18</sup>O data from Fig. 5 (green circles) plotted against their respective label ("lt" stands for Laichinger Tiefenhöhle). The darker the tones of blue, the later the respective measurement was repeated. For interpretation of the references to color in this figure caption, the reader is referred to the web version of this paper.**

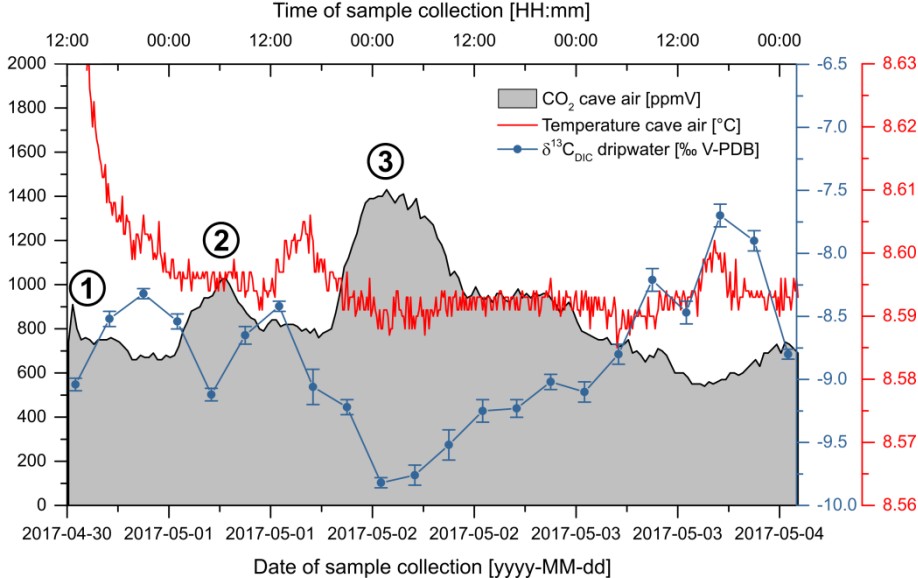

**Fig. 7.** Time series of T, $CO_2$ and $\delta^{13}C_{DIC}$ generated during a first case study applying the GUARD autosampler (units are given in the legend). Cave air $CO_2$ concentrations and dripwater $\delta^{13}C_{DIC}$ values correlate negatively. Main $CO_2$ peaks are highlighted with circled numbers.

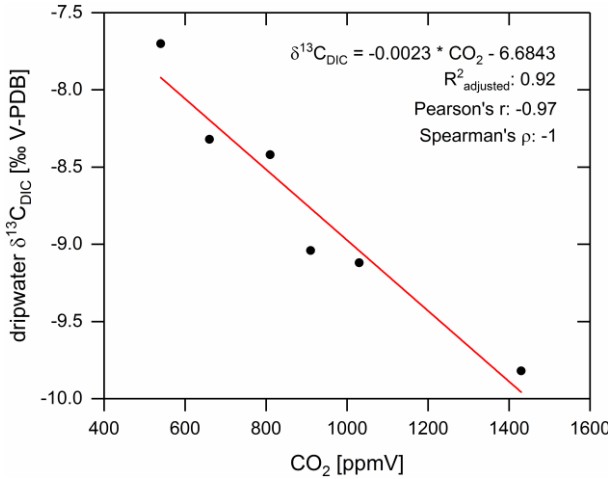

$$\delta^{13}C_{DIC} = -0.0023 * CO_2 - 6.6843$$
$$R^2_{adjusted}: 0.92$$
$$\text{Pearson's } r: -0.97$$
$$\text{Spearman's } \rho: -1$$

**Fig. 8.** Relationship between cave air $CO_2$ concentrations and dripwater $\delta^{13}C_{DIC}$ quantified based on the six maxima/minima recorded during the case study (black circles). The relationship can be fitted very well with a linear regression (red line).

**Tab. 1. Detailed description of the autosampler's integral components.**

| COMPONENTS | DESCRIPTION |
|---|---|
| **Mechanical** | |
| Casing | Peli®, model 1610, heavy-duty, water-tight and airtight, including a valve for automatic pressure purge |
| Z-movement: servo | Reely® Standard RS-610 MG, operating voltage 6.6 V, attached to the Z-slide containing the double-cannula via an elongated hole in the servo's horn |
| X-/Y- movement: motors | Sanyo Denki®, bipolar hybrid stepping motors, 1 A, 24 V, 1.8°/step, 0.265Nm, 4 wires |
| Pump | Peristaltic (flexible-tube) pump, model AP-40; operating voltage 12 V, |
| Sample vials | Labco Exetainer® 738W, soda glass, 12 mL, flat bottom, height (vial + cap) ≤ 101 mm; external ø ≤ 15.5 mm; internal ø ≥ 13.2 mm; including rubber septa with a thickness ≥ 3 mm |
| Tubing | Deutsch & Neumann®, FKM (synthetic rubber, "Viton"), Shore hardness 75, external ø ≤ 6.2 mm, internal ø 4 mm |
| Double cannula | Braun Sterican®, metal, external ø 0.60 mm; length excluding Luer-Lock connector 30 mm |
| **Electronic** | |
| Battery | Panasonic®, valve regulated Pb-acid battery 12 V, 20 Ah, maintenance-free, non-spillable, low self-discharge, 5.8 kg, 76 x 167 x 181 mm; |
| Microcontroller board | Arduino® Mega 2560 including an Atmel ATmega 2560 microcontroller with 54 digital I/O pins, 16 analogue inputs, 6 interrupt inputs, 4 serial interfaces, 1 $I^2C$ interface and 4 KB EEPROM memory (non-volatile); hibernation mode-enabled |
| Real-time clock | RTC PCF8563 powered by a separate 3V lithium button cell battery as a buffer battery |
| Display | Liquid crystal display (LCD) with 2 lines à 16 characters |
| Other electronic components | operating voltage 5 V; 3 DC/DC converters; 2 stepping motor driver carriers: Pololu® A4988; relay board including 2 relays; keypad comprising the characters 1 to 9, * and # |

**Tab. 2. Comparison of the GUARD prototype with conventional autosamplers available on the market, represented here by the 3700C Compact from Teledyne ISCO (information retrieved from the manufacturer's website: www.teledyneisco.com) as well as other non-commercial autosamplers developed by members of the scientific community.**

| Properties | "GUARD" | "3700C Compact" | "OPEnSampler" | "Lisa Liquidsampler" | "Coplen" |
|---|---|---|---|---|---|
| gastight samples | Yes | No | Yes | Yes | (Yes; covered by Teflon lid) |
| sample frequency | 1 min to 168 h | 1 min to 99 h 50 min | 1 min to n.a. | 1 h or 24 h | 30 min or n.a. |
| estimated operating time (at a 2-day interval) | 100 days | < 70 days | n.a. | 100 days | n.a. |
| maximum number of samples vials | 160 | 24 | 24 | 48 | 96 |
| sample volume | ≤ 12 mL | 0.375 to 9.45 L | 250 mL | 20 to 60 mL | 15 mL |
| weight (incl. battery; excluding samples) | 13 kg (+5.8 kg battery) | 11.3 kg | ≥ 25 kg * | ≥ 13 kg (case only) [+] | n.a. |
| weight (incl. battery and samples) | 18.8 to 20.7 kg (depending on sample number) | 15.8 to 23.3 kg (depending on sample size and number) | ≥ 31 kg * | ≥ 14 kg to 16 kg (case and samples; depending on sample size) [+] | ≥ 1.4 kg (samples only) |
| total outer dimensions | H: 31 cm; L: 67.5 cm; W: 53 cm; V: 0.11 m³ | H: 70.5 cm; ø: 45 cm; V: 0.14 m³ | H: 51 cm; L: 109 cm; W: 53 cm; V: 0.29 m³ * | H: 59 cm; L: 80 cm; W: 60 cm; V: 0.28 m³ [+] | n.a. |
| rinse cycle(s) between samples | No | Yes, up to three | Yes | n.a. | No |
| Liquid presence detector (automatic compensation for changes in hydraulic head) | No | Yes | No | No | No |
| Different modes of sample pacing (e.g. time, flow) | No | Yes | No | No | No |

* including the Pelican 80QT Elite Wheeled Cooler for better comparability

[+] including the Zarges K470 Plus 40503 case for better comparability

n.a.: no information found

