# Peer review of "Technical note: GUARD – An automated fluid sampler preventing sample alteration by contamination, evaporation and gas exchange, suitable for remote areas and harsh conditions"

_Hydrology and Earth System Sciences, 2017_

## Referee Comment (RC1) · Anonymous Referee #1 · 1 Feb 2018

The manuscript by Hartman et al. presents a newly-developed automatic fluid sampler (GUARD) that fills the samples into septa-sealed vials to avoid sample alteration due to gas exchange, phase changes, or contamination. The performance of the GUARD system was evaluated with three experiments. In the first experiment, the remaining air in all sample vials was quantified. The authors conclude that the remaining air comprised less than 2% of the total inner volume of the vials and might thus not alter the sample. In a second experiment in a karst cave, the authors compared manually and automatically collected drip water samples and find only minor effects on the $\delta 18O$

values of the samples collected with the two different methods. Lastly, the authors repeatedly measured $\delta18O$ in water samples in nine vials that were initially pierced by the injection needle. The data suggest that the septum of the sampling vials remains air tight over a period of up to six months. Finally, a case study is presented as a practical application of the GUARD system. Drip water from a karst cave was collected over a period of five days at 4h-resolution to measure $\delta13C$ in the liquid water samples. The manuscript finishes with a short interpretation of the case study results and a technical comparison of the GUARD system with another automatic water sampler (3700C Compact from Teledyne ISCO, USA), which is already on the market. The paper is written and organized in a clear way; the quality of the figures is good. I believe that this technical note might be of great interest for the readers of HESS, since an innovative, field-deployable automatic liquid sampling system is presented that potentially allows flexible operation due to low energy consumption and easy handling. Before publication, however, I'd like the authors to address some critical points, which I have outlined below.

General comments: Carry-over effects: The manuscript describes how the sample (12ml) remains in the sampling tube until it is injected into the vial (P3 L27-31). Due to the under-pressure in the tube, a new sample fills the tube when the previous sample leaves it. I'm wondering about the carry-over effects due to the temporary sample storage in the tube, which might be significant, e.g. for instance for streamwater sampling when precipitation events cause drastically changing solute concentrations compared to baseflow conditions. Can you elaborate on potential carry-over effects in the tubing and what could be done about it (e.g., flushing with air or sample water)? If the sampling aims at analyzing organic constituents, biofilm growth inside the tube might alter the sample, especially when the sample interval is long, e.g. several days? What could be done to prevent biofilm growth?

Fractionation effects during sample storage: During the third experiment you conclude that no alteration of the sample occurred because of the constant $\delta18O$ values (Fig.

6). Do you get the same results when using d2H? Since your samples were analyzed with a LGR, both isotopes should be measured simultaneously.

Check-standard during long-term sampling: In the case study, the GUARD system was operating over a period of 5-days and $\delta$13C was measured in the 22 drip water samples. How can you be sure that the $\delta$13C values you have measured were not affected by the sampling process or the storage? In order to quantify drift effects or alterations due to sample processing, it would have been ideal to regularly sample a check-standard with known $\delta$13C in addition to the drip water samples. I would recommend to at least address this issue in the interpretation section of the results.

Harsh conditions: You state that the GUARD system is applicable in harsh (outdoor) conditions (title, P1 L19), which should include a wide range of air temperatures. However, there is no analysis of potential evaporation effects of the samples in very warm (and dry) environments. Instead, during the only long-term experiment that focused on the gas-tightness of the sampling vials, the samples were stored in the fridge at 8°C (P6 L29). In a warm (and dry) environment, I would expect the evaporative fractionation effect to be detectable, especially if the sample sits in the sampling tube for a while before it is injected into the vials. Could you please elaborate on this?

Specific comments:

P6 L10 and Fig. 5: You describe that you have collected one drip sample per day over a period of 33 days, however, in Fig. 5 only 14 data points from the GUARD system are shown, and these are clearly not in daily intervals. Please correctly state the used sampling interval in the text.

P6 L21-26 and Fig. 5: Why don't you show the remaining data points in Fig. 5 to support your claim that the isotopic composition in drip water can vary strongly over short periods? In this context, I would suggest to also provide the standard deviation to the arithmetic mean value in L25. If the standard deviation is substantial (which you suggest with your statement in L21-23), your conclusion based on the arithmetic

means would be invalid.

P6 L2-3: Why didn't you simply weight the vials before and after filling in order to quantify the sample volumes?

P7 L 27: Sampling for 5 days, every 4 hours would yield 30 samples, not 22. What happened to the remaining 8 samples?

Fig. 5: Why are the error bars different for some points? Please indicate in the figure caption, what the errors pars represent (measurement uncertainty?). You should also report d2H values in Fig. 5 since they are measured anyway.

Fig. 6: In greyscale, the shading of the data points is difficult to distinguish (green versus light blue). I would suggest a different way to present these data, especially since some data points overlap with each other and the error bars.

Tab. 2: The sample volume can be smaller than 12ml in the GUARD system.

---

## Referee Comment (RC2) · R. Hut (Referee) · 7 Feb 2018

**Review of "GUARD – An automated fluid sampler preventing sample alteration by contamination, evaporation and gas exchange, suitable for remote areas and harsh conditions" by Arno Hartmann et al.**

Review by dr. ir. Rolf Hut

The authors present a novel autosampler that, as per their claim, seals the sample with the outside world after taking it. They rightly claim that common availability of a sampler like this will greatly help the science of hydrology.

Apart from a view minor suggestions I believe are easy for the authors to take into account, I recommend publishing this article in HESS. On top of that, I would also argue that the readership of the EGU- Copernicus journal Geoscientific Instrumentation[1] (GI) will be interested in this work since I see broader application of this device in the geosciences than just in hydrology.

My comments focus on two issues:
1. Validity of the device and claims made in the article
2. Open Science and reproducibility

**Validity and claims**

The authors claim that their device prevents contact with the environment, including evaporation of the sample, after the sample is taken. They demonstrate in their fieldwork in the karst cave that their samples are statistically identical to manual samples. They furthermore show that their samples do not deteriorate over time by repeating the measurements. The questions I have:
1. I would expect an autosampler to take measurements at regular intervals. However, in figure 5 the samples seem to be taken at rather random times. Can the authors explain why this is?
2. The authors substantiate their claim that the samples are kept airtight by placing them in a fridge for a considerate amount of time. However, there is no control to compare against, ie. no open samples that are exposed to evaporation in that fridge. It is hard for the readership to judge the amount of expected evaporation had the samples not been properly sealed. I would find it unreasonable to ask the authors to redo their experiments, but would like to ask them to provide the readership with an
* * *
[1] Disclaimer: I am not an editor on the GI team, although I did publish in it.

estimate of expected evaporation in the setting of their fridge (8 degrees C, high humidity I guess?) based on literature values. This will help to show that indeed, their samples are sealed properly.

**Open Science and reproducibility**

HESS is a fully Open Access journal and the editors also actively advocate for Open and Reproducible Science in general. In this spirit I think that although the article is it now stands informs the readership about the existence of the new autosampler, it does not allow hydrologists to start using it. The provided technical details are insufficient to rebuild the GUARD using just this article. If the authors intended this (because they maybe want to persue manufacturing the GUARD commercially?) than I think that HESS might not be the ideal outlet to promote it, it is after all a non-for-profit Open Access Scientific journal, not a commercial advertisement leaflet.

I hope the authors did intend the GUARD to be re-buildable by other hydrologists, which would be completely in the spirit of Open Hardware, the movement spearheaded by the Arduino which the authors use as main CPU. By providing a flowchart of their code and their electrical circuitry the authors do hint that this is their intention. For the GUARD to be fully re-buildable I would ask the authors to add:

3. A detailed technical drawing of the physical device, including sizes of all components
4. A Bill of Materials akin to their Table 1, but with more detail. At least the price and an (online?) location where the part can be bought at time of publishing should be included.
5. A step by step build guide. This could be hosted on an external website like instructables.com and linked to in the article, it could also be provided as supplementary material

In this way, the authors will help the readership to have the most benefit from their research.

**Minor points**

6. The opens lab at OSU[2] is also working on an autosampler, with a complete different setup. Might be worth citing their work: http://www.open-sensing.org/opensampler/. They have a paper forthcoming, but did present it at the AGU fall meeting (where I spotted it). Maybe that abstract can be cited.
7. On line 3 of page 3 the terms "high frequency, long term monitoring" etc. are used. What constitutes high or long term is very dependent of the field of science one is in. Please make this more specific to the GUARD.
8. On page 3, line 6: I had to look up what "septa" is. Maybe this is because I'm not a native English speaker. If septa is considered a technical term, please explain it once you introduce it for the first time.
9. On page 4, line 24: future work might be better place in the discussion, although mentioning it at both places is also fine.
* * *
[2] Disclaimer: the director of the opens lab at OSU, prof. John Selker is both a professional and a personal friend of mine.

10. On page 6, line 4: "effectively prevented" assumes certain demands from applications. I suggest with replacing with something like: "prevented for most common use cases".
11. On page 18, table one: sentences like "the sampler can also run… … important constraint" are more suited in the discussion.

Good luck with these final points and finishing this nice publication.

Best wishes,

Dr. ir. Rolf Hut

---

## Short Comment (SC1) · 6 Mar 2018

Hartmann et al. present an automatic battery-operated sampler that takes water samples at pre-programmed time intervals and seals them to prevent atmospheric contact. The suggested method, i.e., the injection of water with a double-cannula into septum-sealed vials (arranged in an X-Y-grid), is rather elegant. Additionally, the number of vials (currently 48, but up to 160) is substantial. Hence, I share the authors' view that the presented device has great potential in hydrology, which warrants publication in HESS.

Nevertheless, there are a few minor points that I would like to mention:

The authors emphasize several times that available autosamplers do not seal collected samples (e.g., page 2, line 13-14; page 2, line 32-33) and selected one commercial device for comparison. Indeed, this sampler (ISCO 3700C Compact) does not prevent atmospheric contact. However, autosamplers that are capable of sealing samples after collection do exist. The following list might not be complete, but these are devices I have stumbled upon in the course of my own literature review (disclaimer: I am currently involved in the design and testing of an automatic rain collector):

1. OPEnSampler by OPEnS Lab (http://www.open-sensing.org/opensampler/; see review by Rolf Hut)

2. Lisa Liquidsampler by Lukas Neuhaus (https://www.liquidsampler.de/)

3. Sequential, time-integrating precipitation collector by Coplen et al. (2008; see Supporting Information)

The first two devices have apparently not been formally published and the second website is currently only available in German. Although it may be quite easy to miss these models in a literature review, they do exist and I would like to suggest that they be mentioned in the paper for the sake of completeness. Including them will not diminish the value of the authors' contribution. Although there are a few other devices (with somewhat different specifications), the sampler by Hartmann et al. is still a useful addition to those already in existence, particularly if presented in a way that enables reproduction (see review by Rolf Hut).

Additionally, the section on potential applications attracted my attention. I am a bit confused about the authors' idea to use their sampler in the Global Network of Isotopes in Precipitation (GNIP; see Section 5). Currently, it sounds as if they suggest replacing the current cumulative collectors with their automatic sampler. As far as I know, the main aim of GNIP is to collect integral samples, i.e., samples that represent the entire

precipitation occurring during the collection period (usually a month). The samples are then routinely analyzed for $\delta$18O, $\delta$2H, and partly 3H. I am not sure how this could be achieved with the model described in the manuscript. In the current setup, one "collected sample represents the water under investigation at a given instant (integrated over 22 seconds)" (page 4, line 15-16). Maybe the authors could provide more details on the potential deployment as part of GNIP. Would they still use a peristaltic pump or would the rainwater flow into the vials by gravity? Would they use the same vial number (48) and size (12 mL)? How would they approach programming collection intervals, without knowing when it will rain? Could their sampler also be used at GNIP sites exhibiting harsh conditions (i.e., a warm and arid climate)? Alternatively, the authors could phrase their idea more carefully, for example by suggesting the addition of their device to the cumulative collectors at GNIP stations (instead of replacing them).

I hope these minor comments are helpful and perhaps contribute to further improvement of the manuscript, which is already a good contribution in presenting a useful automatic sampler.

Best regards,

Nils Michelsen

References: Coplen, T. B., Neiman, P. J., White, A. B., Landwehr, J. M., Ralph, M., and Dettinger, M. D.: Extreme changes in stable hydrogen isotopes and precipitation characteristics in a landfalling Pacific storm, Geophys. Res. Lett., 35, L21808, 2008.
* * *

---

## Author Comment (AC1) · 4 Apr 2018

**Authors' response to referee comment 1**

**General comments:**

**Referee Comment**: Carry-over effects: The manuscript describes how the sample (12ml) remains in the sampling tube until it is injected into the vial (P3 L27-31). Due to the under-pressure in the tube, a new sample fills the tube when the previous sample leaves it. I'm wondering about the carry-over effects due to the temporary sample storage in the tube, which might be significant, e.g. for instance for streamwater sampling when precipitation events cause drastically changing solute concentrations compared to baseflow conditions. Can you elaborate on potential carry-over effects in the tubing and what could be done about it (e.g., flushing with air or sample water)?

**Authors' response**: Carry-over effects might occur with the device setup as presented in our paper, in particular, as referee #1 points out, when the chemical composition varies strongly between consecutive samples. Carry-over effects could be effectively prevented by thoroughly flushing the tubing with sample water, either prior to sample pre-collection, or prior to sample injection. If such a flushing step is implemented, sample pre-collection becomes obsolete. During development of the presented device we regarded minimizing both the power-consumption and the technical complexity as a higher-priority requirement than preventing carry-over effects through flushing. However, a flushing step could still be implemented without the need of any fundamental changes to the current system. It is important to bear in mind that, in some sampling scenarios, flushing is not a viable option, especially in scenarios where the sample water is not provided in sufficient quantity or continuity, for example during rainwater or cave dripwater sampling. In sampling scenarios focussing on water isotopes, carry-over effects are likely to be minor as the water molecules to be analysed for oxygen isotope composition do not strongly bond to the tubing's wall, but are readily flushed out of the tubing during sample injection. Furthermore, the isotopic composition of natural waters is unlikely to change drastically between consecutive samples.

**Referee Comment**: If the sampling aims at analysing organic constituents, biofilm growth inside the tube might alter the sample, especially when the sample interval is long, e.g. several days? What could be done to prevent biofilm growth?

**Authors' response**: As the tubing is contained within a sealed case protecting the tubing and sample vials from sunlight, the probability of biofilm growth is already diminished compared to a system exposed to light. As some microorganisms are capable of forming biofilms in the absence of light, to further prevent the formation of biofilms, antimicrobial coatings could be applied to the inner walls of the tubing, such as antibiotics, biocides or colloidal silver coatings that are commonly used on medical devices to prevent infection (e.g. Ramasamy & Lee, 2016). The most practical solution to the potential problem of biofilm growth is probably the use of silver plated metal tubing instead of the FKM tubing presented in the paper.

**Referee Comment**: Fractionation effects during sample storage: During the third experiment you conclude that no alteration of the sample occurred because of the constant $\delta^{18}O$ values (Fig. 6). Do you get the same results when using $d^2H$? Since your samples were analysed with a LGR, both isotopes should be measured simultaneously.

**Authors' response**: Yes, the δD results (see Fig. 6b) also confirm the long-term stability of the samples: Again, if the vials were not airtight, evaporation would have led to a preferential removal of isotopically light water molecules from the water samples due to their higher vapour pressure (e.g.

Hoefs, 2015) and, consequently, to an increase of the δD value of the remaining water sample over time. Such a positive trend is not present in the δD data and the results from the repeated measurements agree well with the initial ones. The difference in δD values between initial and repeated measurements ranges from -0.30 ‰ (lt20 and lt23) to 0.70 ‰ (lt02-05), but averages out at 0.0 ‰ over all measurements (median also 0.0 ‰) indicating that there is no systemic discrepancy between initial and repeated analyses (Fig. 6b).

[Figure]

**Fig. 6b: Results of repeated δD measurements (circles in tones of blue) measured in the automatically collected samples together with the original δD data from Fig. 5 (green circles) plotted against their respective label ("lt" stands for Laichinger Tiefenhöhle). The darker the tones of blue, the later the respective measurement was repeated.**

**Changes to the manuscript**: Insert Fig. 6b in the Supplements and include a reference to Fig. 6b in the text.

**Referee Comment**: Check-standard during long-term sampling: In the case study, the GUARD system was operating over a period of 5-days and $\delta^{13}$C was measured in the 22 drip water samples. How can you be sure that the $\delta^{13}$C values you have measured were not affected by the sampling process or the storage? In order to quantify drift effects or alterations due to sample processing, it would have been ideal to regularly sample a check-standard with known $\delta^{13}$C in addition to the drip water samples. I would recommend to at least address this issue in the interpretation section of the results.

**Authors' response**: The purpose of the case study performed in the cave "Kleine Teufelshöhle" was to monitor the changes in dripwater $\delta^{13}C_{DIC}$ values with varying cave $pCO_2$ after the dripwater had equilibrated with the cave atmosphere via $CO_2$ degassing. Therefore, if potential drift effects or alterations in dripwater $\delta^{13}C_{DIC}$ values caused by sample processing were to be examined using a check-standard of known $\delta^{13}C_{DIC}$, this standard would have to be treated exactly as the sampled dripwater, i.e. allowed to degas prior to sampling. This would however alter the $\delta^{13}C_{DIC}$ value of the standard, depending on the varying $pCO_2$ difference between dripwater and cave atmosphere, thus inevitably hampering the use of the check-standard as a control with known $\delta^{13}C_{DIC}$. However, an aliquot of the $CO_2$-equilibrated check-standard could be sampled manually and injected into an airtight sample vial with a double-cannula syringe, shortly before another aliquot of the check-standard is collected automatically by the GUARD autosampler. Comparison of the $\delta^{13}C_{DIC}$ values of

both "samples" should enable for detecting any potential sample alterations during automatic sampling. In agreement with the comment of referee #1, we will address the issue of potential sample alterations in section 4.4.

**Changes to the manuscript**: Insert at the end of section 4.4: "We note that potential drift effects or sample alterations that might be caused by the automatic sampling process have not yet been examined in detail. Corresponding tests using check-standards of known $\delta^{13}C_{DIC}$ values will be performed in future studies."

**Referee Comment**: Harsh conditions: You state that the GUARD system is applicable in harsh (outdoor) conditions (title, P1 L19), which should include a wide range of air temperatures. However, there is no analysis of potential evaporation effects of the samples in very warm (and dry) environments. Instead, during the only long-term experiment that focused on the gas-tightness of the sampling vials, the samples were stored in the fridge at 8°C (P6 L29). In a warm (and dry) environment, I would expect the evaporative fractionation effect to be detectable, especially if the sample sits in the sampling tube for a while before it is injected into the vials. Could you please elaborate on this?

**Authors' response**: The statement that the GUARD autosampler is applicable under harsh conditions mainly refers to its rugged water-tight casing and its ability to prevent damage from extreme weather conditions (e.g. water or dust ingress, high humidity, etc.) and to protect the samples from any external interference, e.g. from animal activity. However, this statement can be expanded to include the samples, too: Once, the samples are injected in the airtight vials, evaporative fractionation as well as other forms of sample alteration are effectively prevented, regardless of ambient air temperature or temperature fluctuations. It is certainly true that the sample is most prone to change during pre-storage in the FKM tubing. During this phase of the sampling, evaporative fractionation is at least minimised through two mechanisms: First, the FKM tubing is highly impermeable to gases and thus impedes evaporation and/or gas exchange through its walls. Second, evaporation can only occur over a very small surface of only about 12.6 mm$^2$ thanks to the small inner diameter of the tubing of only 4 mm. Furthermore, sample pre-storage inside the tubing is not necessary if sample water is provided in sufficient quantity and continuity, for instance, when sampling water from rivers, lakes or the ocean. In these cases, the sample can be injected directly into the sample vial and is therefore almost instantly sealed from the surrounding atmosphere.

**Specific comments:**

**Referee Comment**: P6 L10 and Fig. 5: You describe that you have collected one drip sample per day over a period of 33 days, however, in Fig. 5 only 14 data points from the GUARD system are shown, and these are clearly not in daily intervals. Please correctly state the used sampling interval in the text.

**Authors' response**: The sampling interval is correctly stated as daily, however, not all of the 33 samples were analysed for $\delta^{18}O$ values.

**Changes to the manuscript**: Add at the end of the caption to Fig. 5: "Not all of the 33 samples were analysed for isotopic composition."

**Referee Comment**: P6 L21-26 and Fig. 5: Why don't you show the remaining data points in Fig. 5 to support your claim that the isotopic composition in drip water can vary strongly over short periods?

In this context, I would suggest to also provide the standard deviation to the arithmetic mean value in L25. If the standard deviation is substantial (which you suggest with your statement in L21-23), your conclusion based on the arithmetic means would be invalid.

**Authors' response**: While all of the manually collected samples were analysed for $\delta^{18}O$ values, not all of the hourly samples collected by the autosampler were measured. However, the sum of 16 samples over a period of 26.5 hours is sufficient to establish that there is a certain variation in dripwater $\delta^{18}O$ values on time scales as short as 30 minutes. The (absolute) standard deviation for the 14 automatically collected samples is 0.07 ‰ and 0.06 ‰ for the 12 manually collected samples. Based on the small difference of only 0.03 ‰ between the arithmetic mean $\delta^{18}O$ values calculated for both sample types, we concluded that there is no systematic discrepancy between the automatically and the manually collected samples. This conclusion holds true even if dripwater $\delta^{18}O$ values vary on time scales as short as 30 minutes as this variation includes both positive and negative excursions from the long-term mean values.

**Referee Comment**: P6 L2-3: Why didn't you simply weight the vials before and after filling in order to quantify the sample volumes?

**Authors' response**: Weighing the vials before and after sample injection is another way of quantifying the sampled volumes. As the sample vials were almost entirely filled during the various test runs we conducted, quantifying the sampled volumes by means other than visually confirming that only small air bubbles remained after sample injected simply did not seem necessary.

**Referee Comment**: P7 L 27: Sampling for 5 days, every 4 hours would yield 30 samples, not 22. What happened to the remaining 8 samples?

**Authors' response**: The remaining 8 samples could not be successfully collected during the case study due to an imprecise positioning of the sample slide and double-cannula at the position of sample 24. We have already been able to trace this positioning error to a faulty motor driver. We have therefore installed a new motor driver and achieved both precise and reliable positioning results since this change.

**Referee Comment**: Fig. 5: Why are the error bars different for some points? Please indicate in the figure caption, what the errors pars represent (measurement uncertainty?). You should also report d2H values in Fig. 5 since they are measured anyway.

**Authors' response**: The error bars represent the precision of each individual measurement. It includes the precision of the ten internal sweeps performed by the mass spectrometer on a single sample and the precision of multiple (two to three) measurements of the same sample. The error is propagated using the formula x= (a^2+b^2)^0.5, with x being the propagated error and a and b representing the two error types outlined above.

**Changes to the manuscript**: We will include the measured $\delta D$ values in Fig. 5.

**Referee Comment**: Fig. 6: In greyscale, the shading of the data points is difficult to distinguish (green versus light blue). I would suggest a different way to present these data, especially since some data points overlap with each other and the error bars.

**Authors' response**: We have changed Fig. 6 so that the data points do not overlap any more. As 7 different measurement dates need to be illustrated in this figure, indicating the different

measurement dates with different data point symbols or shadings is neither practical nor intuitive in this case. For suggest to include in the figure's caption "For interpretation of the references to color in this figure caption, the reader is referred to the web version of this paper."

**Referee Comment**: Tab. 2: The sample volume can be smaller than 12ml in the GUARD system.

**Authors' response**: That is correct. The sample volume can be defined by changing the duration of the pumping step during sampling. Headspace is minimal if the vials are filled to the maximum.

**Changes to the manuscript**: We will include a ≤ sign in Tab. 2.

**References**

Ramasamy, Mohankandhasamy; Lee, Jintae (2016): Recent Nanotechnology Approaches for Prevention and Treatment of Biofilm-Associated Infections on Medical Devices. In: *BioMed research international* 2016, S. 1851242. DOI: 10.1155/2016/1851242.

---

## Author Comment (AC2) · 4 Apr 2018

**Authors' response to referee comment 2**

**Referee Comment**:

I.     **Validity and claims:**

The authors claim that their device prevents contact with the environment, including evaporation of the sample, after the sample is taken. They demonstrate in their fieldwork in the karst cave that their samples are statistically identical to manual samples. They furthermore show that their samples do not deteriorate over time by repeating the measurements. The question I have:

1.  I would expect an autosampler to take measurements at regular intervals. However, in Figure 5 the samples seem to be taken at rather random times. Can the authors explain why this is?

**Authors' response**: The karst dripwater samples for which Fig. 5 shows the $\delta^{18}O$ values, have been collected automatically using the GUARD autosampler at (regular) hourly intervals from 13:30 o'clock on December 12, 2016 to 07:30 o'clock on December 13, 2016 and from 12:00 o'clock to 15:00 o'clock and at daily intervals from December 13, 2016 to January 14, 2017. However, not all of the collected samples were analysed for isotopic composition. This is why the samples seem to be unevenly spaced in time. As referee #1 deduced an incorrectly stated sampling interval, we will mention at the end of the caption to Fig. 5 that not all of the collected samples were also analysed.

**Changes to the manuscript**: Add at the end of the caption to Fig. 5: "Not all of the 33 samples were analysed for isotopic composition."

2.  The authors substantiate their claim that the samples are kept airtight by placing them in a fridge for a considerable amount of time. However, there is no control to compare against, i.e. no open samples that are exposed to evaporation in that fridge. It is hard for the readership to judge the amount of expected evaporation had the samples not been properly sealed. I would find it unreasonable to ask the authors to redo their experiments, but would like to ask them to provide the readership with an estimate of expected evaporation in the setting of their fridge (8 °C, high humidity I guess?) based on literature values. This will help to show that indeed, their samples are sealed properly.

**Authors' response**:

We agree with referee #2 that we could have demonstrated the airtightness of the sample vials after sample injection even better had we implemented control samples that are not entirely sealed from the atmosphere and thus exposed to evaporation. To compensate for that caveat and to provide the readership with a notion of the effect of evaporation on the sample $\delta^{18}O$ values, we have calculated both evaporation and $\delta^{18}O$ change for the conditions prevalent in our fridge. Despite being set to 8 °C, the temperature in the fridge was measured to be 11.2 °C, relative humidity was 24 % according to measurements. Based on these conditions and assuming an opening of the sample vial of 5 % to imitate a minor lack of airtightness, evaporation was calculated using a formula that has proven adequate for inactive indoor swimming pools that are not influenced by direct sunlight or wind (Smith, Löf and Jones, 1994) using a water density of 1 g/cm$^3$:

$$\frac{\dot{m}}{A} = \frac{(30.6 + 32.1 * v_w)(P_w - P_a)}{\Delta H_v}$$

where $\dot{m}/A$ is the evaporation rate [kg/(m$^2$ hr)], $v_w$ is the air velocity over the water surface [m/s],

$P_w$ is the saturation vapour pressure at the water temperature [mm Hg], $P_a$ is the saturation vapour pressure at the air dew point [mm Hg] and $\Delta H_v$ is the latent heat of water at the pool temperature [kJ kg].

The $\delta^{18}O$ value of the residual water remaining at each given time was calculated on the basis of a fractionation factor $\alpha$ between water and vapour according to the following formula (e.g. Clark and Fritz, 1999):

$$\, ln \propto_{water-vapour} = 1.137\left(10^6/T_k^2\right) - 0.4156(10^3/T_k) - 2.0667$$

where $T_k$ represents the temperature of the phase change [K] and on the following relationship (e.g. Hoefs, 2015):

$$\frac{R_w}{R_{w0}} = f^{(\frac{1}{\propto}-1)}$$

where $R_w$ is the isotope ratio of the water at a given time [‰ V-SMOW], $R_{w0}$ is the initial isotope ratio of the water [‰ V-SMOW], and f is the fraction of the residual water [-]. The results of these calculations (Fig. 1) demonstrate that even a small slit in a sample vial's rubber septum equalling only 5 % of the vial's inner cross section leads to a substantial shift towards higher $\delta^{18}O$ values in the residual water over time. After three months (90 days), for instance, $\delta^{18}O$ values have risen from -10.1 ‰ by about 1.3 ‰ to -8.8 ‰. The difference between the lowest and the highest $\delta^{18}O$ value in Fig. 6 of the manuscript is still below 0.3 ‰, while those data points span a longer period of six months. Most importantly, there is no positive trend in the $\delta^{18}O$ values in Fig. 6 of the manuscript which illustrates the sample vials are sealed properly, even after sample injection.

[Figure]

Fig. 1: Effect of evaporation on the $\delta^{18}O$ value of the residual water in a 12 mL sample vial at a temperature of 11.2 °C and a relative humidity of 24 %.

**Changes to the manuscript**: We will add Fig. 1 in the authors' response to the Supplementaries including the corresponding explanations as above. In the manuscript, we will a shortened version of these explanations at the end of Section 3.3 (page 7, line 3).

**Referee Comment**:

**II.  Open Science and reproducibility:**

HESS is a fully Open Access journal and the editors also actively advocate for Open and Reproducible Science in general. In this spirit I think that although the article as it now stands informs the readership about the existence of the new autosampler, it does not allow hydrologists to start using it. The provided technical details are insufficient to rebuild the GUARD using just this article. If the authors intended this (because they maybe want to pursue manufacturing the GUARD commercially?) then I think that HESS might not be the ideal outlet to promote it, it is after all a non-for-profit Open Access Scientific journal, not a commercial advertisement leaflet.

I hope the authors did intend the GUARD to be re-buildable by other hydrologists, which would be completely in the spirit of Open Hardware, the movement spearheaded by the Arduino which the authors use as main CPU. By providing a flowchart of their code and their electrical circuitry the authors do hint that this is their intention. For the GUARD to be fully re-buildable I would ask the authors to add:

1. A detailed technical drawing of the physical device, including sizes of all components

**Authors' response**: Any user of the GUARD autosampler would benefit from it the most if the sampler's dimensions are precisely adapted to the user's specific requirements. Therefore, the exact dimensions of the GUARD device presented in the manuscript are not relevant as the device dimensions should be regarded flexible rather than fixed. The necessary minimum dimensions mainly depend on the number of sample vials needed. In the setup presented in the manuscript the GUARD autosampler comprises 48 sample vials, but can be equipped with up to 160 sample vials at the given casing dimensions by maximising the dimensions of the sample rack (to the left side in Fig. 1 in the manuscript) and by reducing the space between adjacent sample vials to zero. If a higher number of sample vials is needed, the sampler dimensions need to be adapted accordingly. Only few components of relevant size have fixed dimensions (Table 1). All x-profiles and guides that form the framework within which the x- and y-slides move have to be cut to fit into the chosen casing.

Table 1: Integral components of the GUARD autosampler of relevant size

| COMPONENTS | Description | Dimensions |
|---|---|---|
| **Mechanical** | | Length x Width x Height |
| Z-movement: servo | Reely® Standard RS-610 MG, operating voltage 6.6 V, attached to the Z-slide containing the double-cannula via an elongated hole in the servo's horn | 40x20x42 mm |
| X-/Y- movement: motors | Sanyo Denki®, bipolar hybrid stepping motors, 1 A, 24 V, 1.8°/step, 0.265Nm, 4 wires | 42x42x24 mm |
| Pump | Peristaltic (flexible-tube) pump, model AP-40; operating voltage 12 V, | 55x50x42 mm |

**Changes to the manuscript**: Add Table 1 in the autors' response to the Supplementaries.

2. A Bill of Materials akin to their Table 1, but with more detail. At least the price and an (online?) location where the parts can be bought at the time of publishing should be included.

**Authors' response / Changes to the manuscript**: We will add the following Bill of Materials to the Supplementaries

| Components | Description | quantity | cost/unit | total cost | company | purchase order no. |
|---|---|---|---|---|---|---|
| **Mechanical** | | | | | | |
| Casing | Peli®, model 1610, heavy-duty, water-tight and airtight, including a valve for automatic pressure purge | 1 | 252.35 € | 252.35 € | Waterproof-Cases | - |
| Z-movement: servo | Reely® Standard RS-610 MG, operating voltage 6.6 V, attached to the Z-slide containing the double-cannula via an elongated hole in the servo's horn | 1 | 12.60 € | 12.60 € | Conrad Electronic | 1365925 - 05 |
| X-/Y- movement: motors | Sanyo Denki®, bipolar hybrid stepping motors, 1 A, 24 V, 1.8°/step, 0.265Nm, 4 wires | 2 | 38.95 € | 77.90 € | RS Components | 829-3499 |
| Pump | Peristaltic (flexible-tube) pump, model AP-40; operating voltage 12 V, | 1 | 19.90 € | 19.90 € | Gemke Technik GmbH | APE40CD12V |
| Sample vials | Labco Exetainer® 738W, soda glass, 12 mL, flat bottom, height (vial + cap) ≤ 101 mm; external ø ≤ 15.5 mm; internal ø ≥ 13.2 mm; including rubber septa with a thickness ≥ 3 mm; 48 vials of 300 in a packaging unit | 1 | 22.28 | 22.28 € | IVA | IVA738W |
| Tubing | Deutsch & Neumann®, FKM (synthetic rubber, "Viton"), Shore hardness 75, external ø ≤ 6.2 mm, internal ø 4 mm | 3 | 12.90 € | 38.70 € | häberle Shop | 9.205 765 |
| Double cannula | Braun Sterican®, metal, external ø 0.60 mm; length excluding Luer-Lock connector 30 mm | 2 | 3.40 € | 6.80 € | häberle Shop | 7.079 505 |
| Framework for slide movement | Makeblock XY Printer | 1 | 269.95 € | 269.95 € | Eckstein | MB90014 |
| **Electronic** | | | | | | |
| Battery | Panasonic®, valve regulated Pb-acid battery 12 V, 20 Ah, maintenance-free, non-spillable, low self-discharge, 5.8 kg, 76 x 167 x 181 mm; the sampler can also run on 12 V Li-ion batteries if weight is an important constraint | 1 | 75.03 € | 75.03 € | Voelkner | S167901 |
| Microcontroller board | Arduino® Mega 2560 including an Atmel ATmega 2560 microcontroller with 54 digital I/O pins, 16 analogue inputs, 6 interrupt inputs, 4 serial interfaces, 1 I$^2$C interface and 4 KB EEPROM memory (non-volatile); hibernation mode-enabled | 1 | 21.99 € | 21.99 € | Conrad | 1409778 - 05 |
| Real-time clock | RTC PCF8563 powered by a separate 3V lithium button cell battery as a buffer battery | 1 | 10.91 € | 10.91 € | Conrad | 1195070 - 05 |
| Display | Liquid crystal display (LCD) with 2 lines à 16 characters | 1 | 9.87 € | 9.87 € | Conrad | 183045 - 05 |
| **Other electronic components:** | relay module | 1 | 8.52 € | 8.52 € | Exptech | EXP-R25-187 |
| | drivers for stepping motors | 2 | 7.95 € | 15.90 € | Exptech | EXP-R25-001 |
| | casing for control panel | 1 | 5.28 € | 5.28 € | Conrad | 522641-99 |
| | DC/ DC converter 12V | 1 | 12.00 € | 12.00 € | Conrad | 154170-05 |
| | DC/ DC converter 5V | 1 | 2.65 € | 2.65 € | Conrad | 157954-05 |
| | DC/ DC converter 6,5V | 1 | 5.82 € | 5.82 € | Conrad | 156674-05 |
| | CR2032 3V lithium button cell battery as a buffer battery | 1 | 2.26 € | 2.26 € | Conrad | 1086225-05 |
| | USB service interface FrontCom® Micro IE-FCM-USB-A Weidmüller | 1 | 20.35 € | 20.35 € | Conrad | 746885-05 |
| | Membrane keypad Matrix 1 x 12 SU709948 | 1 | 11.11 € | 11.11 € | Conrad | 1341283-62 |
| | 3D print-outs (sample rack, connectors, double-canula adapter) | 1 | 15.00 € | 15.00 € | - | - |
| | Aluminium slot profiles 20x20 mm Slot 5 (m) | 1 | 2.94 € | 2.94 € | Motedis | 19586 |
| | Sliding nuts Slot 5 100 pieces | 1 | 21.42 € | 21.42 € | Motedis | 96214 |
| | Screw  DIN 7984 M4x10 Slot 5 | 100 | 0.12 € | 12.00 € | Motedis | - |
| | Bracket 20x40 I-type Slot 5 10 pieces | 3 | 7.50 € | 22.50 € | Motedis | 093W202N05 |
| | Swivel Feet. Series 10 PA; foot 40, threaded rod 5x60 4 pieces | 4 | 1.00 € | 4.00 € | Motedis | - |
| | Miniature sliding rail IGUS drylin TK-04 | 1 | 10.16 € | 10.16 € | IGUS | TS-04-07 |
| | CNC Aluminium Servo Horn 60mm for Futaba servos 25 teeth | 1 | 6.90 € | 6.90 € | Ebay | 251439671553 |
| | Cable gland PG7 Polyamide black (RAL 9005) KSS EGRWW7 water-tight | 1 | 0.34 € | 0.34 € | Conrad | 533738-05 |
| | zip ties different sizes 200 pieces | 1 | 3.80 € | 3.80 € | Conrad | 541665-62 |
| | USB cable PC/Sampler | 1 | 4.29 € | 4.29 € | Conrad | 1592198-62 |
| | Merck® silicone grease for sealing 100gr. | 1 | 68.70 € | 68.70 € | häberle Shop | 1.07746.0100 |
| | Hose fitting, straight, 4040 | 10 | 2.15 € | 21.50 € | häberle Shop | 9.207 801 |
| | **Total** | | | **1,095.72 €** | | |

3. A step by step build guide. This could be hosted on an external website like instructables.com and linked to in the article, it could also be provided as supplementary material

**Authors' response**: We find the idea of referee #2 of a step-by-step build guide highly intriguing and would like to provide such a guide in the near future, being aware of the potential benefit it might have for researchers and other users who want to build their own GUARD autosampler. However, considering the complex process required building the GUARD autosampler with its many steps and including multiple custom adaptions involving 3D-printouts, we hope that the referees and the editors of HESS understand that a complete step-by-step build guide is clearly beyond the scope of this journal article, even for the Supplementaries. As we certainly do intend to make the GUARD autosampler available to potential users in research and other fields and bearing in mind the complexity of the building process, however, we would like to offer these users our advice during their building process until a step-by-step guide can be provided. As another way of making the GUARD autosampler accessible for the scientific community and other groups, we might lend our device to interested users, of course free of charge. We hope that the referees and the editors of HESS can accept one or both of the offered solutions.

**Minor points:**

**Referee Comment**: The opens lab at OSU is also working on an autosampler, with a completely different setup. Might be worth citing their work: http://www.open-sensing.org/opensampler/. They have a paper forthcoming, but did present it at the AGU fall meeting (where I spotted it). Maybe that abstract can be cited.

**Authors' response**: We will mention the OPEnSampler and cite the conference abstract, as suggested.

**Changes to the manuscript**: Insert at the end of page 2: "(Note that similar types of autosamplers are currently worked on, for instance, the "OPEnSampler" developed at Oregon State University (Nelke, Selker and Udell, 2017)".

**Referee Comment**: On line 3 of page 3 the terms "high frequency, long term monitoring" etc. are used. What constitutes high frequency of long term is very dependent on the field of science one is in. Please make this more specific to the GUARD.

**Authors' response**: We will specify the time scale referred to in the manuscript.

**Changes to the manuscript**: On page 3 in line 3 replace line by "either high-frequency sampling (e.g. every minute), long-term monitoring (e.g. 6 months), or medium-term monitoring at medium sampling frequency (e.g. daily sampling for 48 days)."

**Referee Comment**: On page 3, line 6: I had to look up what "septa" is. Maybe this is because I'm not a native English speaker. If septa is considered a technical term, please explain it once you introduce it for the first time.

**Authors' response**: We will define the term upon its first appearance in the manuscript.

**Changes to the manuscript**: On page 3, in line 6 insert after "septa": "(engineered membranes that permit the transfer of fluids without air contact, usually using a double-canula)"

**Referee Comment**: On page 4, line 24: Future work might be better placed in the discussion, although mentioning it at both places is also fine.

**Authors' response**: Connection multiple batteries in parallel, replacing discharged batteries using an electrical bypass and operating the GUARD autosampler on mains power using an appropriate rectifier are options viable for any battery-powered autosampler. Therefore, these options mentioned on page 4, in line 24, do not distinguish the GUARD autosampler from any other autosampler such as the 3700C Compact from Teledyne Isco. For that reason, we think that mentioning these options again in the discussion (Section 5) would be redundant and inappropriate.

**Referee Comment**: On page 6, line 4: "effectively prevented" assumes certain demands from applications. I suggest replacing it with something like: "prevented for most common use cases".

**Authors' response**: As there might be applications we might not have considered, we agree to change the passage.

**Changes to the manuscript**: On page 6, in line 4 delete "effectively".

**Referee Comment**: On page 18, table one: Sentences like "the sampler can also run … … important constraint" are more suited in the discussion.

**Authors' response**: We agree with referee #2 and will move the indicated sentence.

**Changes to the manuscript**: On page 18, in Table 1 in line "Battery" remove the remark "the sampler can also run on 12 V Li-ion batteries if weight is an important constraint" and insert it on page 4 in line 29 at the end of the paragraph.

**References**

Clark, Ian D.; Fritz, Peter (1999): Environmental isotopes in hydrogeology. [2. print., corr.]. Boca Raton: Lewis Publ.

Hoefs, Jochen (2015): Stable isotope geochemistry. 7. ed. Cham: Springer (Earth Sciences).

Nelke, Mitch; Selker, John, S.; Udell, Chet (2017), The OPEnSampler: A Low-Cost, Low-Weight, Customizable and Modular Open Source 24-Unit Automatic Water Sampler, Abstract H41J-1596 presented at 2017 Fall Meeting, AGU, New Orleans, LA, 11-15 Dec.

Smith, Charles, C.; Löf, George; Jones, Randy (1994): Measurement and analysis of evaporation from an inactive outdoor swimming pool. In: Pergamon 1994, pp. 3-7.

---

## Author Comment (AC3) · 8 Apr 2018

**Authors' response to short comment 1**

**Reader Comment**:

Hartmann et al. present an automatic battery-operated sampler that takes water samples at pre-programmed time intervals and seals them to prevent atmospheric contact. The suggested method, i.e., the injection of water with a double-cannula into septum-sealed vials (arranged in an X-Y-grid), is rather elegant. Additionally, the number of vials (currently 48, but up to 160) is substantial. Hence, I share the authors' view that the presented device has great potential in hydrology, which warrants publication in HESS.

Nevertheless, there are a few minor points that I would like to mention:

The authors emphasize several times that available autosamplers do not seal collected samples (e.g., page 2, line 13-14; page 2, line 32-33) and selected one commercial device for comparison. Indeed, this sampler (ISCO 3700C Compact) does not prevent atmospheric contact. However, autosamplers that are capable of sealing samples after collection do exist. The following list might not be complete, but these are devices I have stumbled upon in the course of my own literature review (disclaimer: I am currently involved in the design and testing of an automatic rain collector):

1. OPEnSampler by OPEnS Lab (http://www.open-sensing.org/opensampler/; see review by Rolf Hut)
2. Lisa Liquidsampler by Lukas Neuhaus (https://www.liquidsampler.de/)
3. Sequential, time-integrating precipitation collector by Coplen et al. (2008; see Supporting Information)

The first two devices have apparently not been formally published and the second website is currently only available in German. Although it may be quite easy to miss these models in a literature review, they do exist and I would like to suggest that they be mentioned in the paper for the sake of completeness. Including them will not diminish the value of the authors' contribution. Although there are a few other devices (with somewhat different specifications), the sampler by Hartmann et al. is still a useful addition to those already in existence, particularly if presented in a way that enables reproduction (see review by Rolf Hut).

**Authors' response**: We thank Mr. Michelsen for notifying us of the existence of other similar liquid autosamplers being developed by other groups. Indeed, we seem to have missed these devices in our own literature research. In response to the comment of Rolf Hut (referee #2) with regard to the OPEnSampler we have already suggested to change the manuscript and to mention and to cite the sampler in our manuscript. However, in view of the fact that the OPEnSampler is not the only device similar to the GUARD autosampler, we suggest a more comprehensive change to the manuscript in order to give appropriate credit to other groups for their sampler developments.

As the mentioned liquid autosampler prototypes are not (yet) available on the market, we argue that our comparison of the GUARD autosampler with the commercially available autosampler 3700C Compact (Teledyne ISCO, USA) in Section 5 of the manuscript is still relevant and thus advocate not omitting this comparison from the manuscript.

**Changes to the manuscript**: On page 2 in line 31, insert "Furthermore, the need for automated liquid sampling in general is demonstrated by a number of technical developments by multiple groups with the aim of creating automated liquid samplers capable of sealing the samples after collection. For instance, researchers at Oregon State University have developed the "OPEnSampler" (Nelke, Selker and Udell, 2017; http://www.open-sensing.org/opensampler/) that comprises an array of 24 solenoid valves, allowing the 24 sampling containers to be sealed from the environment after sample

collection. Lukas Neuhaus has developed the "Lisa Liquidsampler" (not published) that fills 48 sample vials sealed by septa (engineered membranes that permit the transfer of fluids without air contact, usually using a double-canula) using a vacuum pump via 48 separate transfer tubes. Applying a new automated precipitation collector obtaining 96 sequential 15-mL samples, Coplen et al. (2008) were able to measure a strong decrease of 51% in the hydrogen isotope ratio (δD) of precipitation over only one hour resulting from the landfall of an extratropical cyclone along the coast of California. Evaporation and subsequent isotopic fractionation was minimised by a Teflon-coated vial cover, thus sample vials are not sealed individually."

To harmonise the rest of the manuscript with this change, we suggest to replace the sentence on page 2 in line 32 with "In addition to these newly developed liquid autosamplers that are 1) suited for field operation in remote areas and under harsh (outdoor) conditions and 2) capable of sealing the sample vials (gastight) directly after sample collection, we have designed, constructed and tested a new autosampler ("GUARD") that also fulfils these requirements, but can be equipped with up to 160 sample vials due to its space-efficient design."

**Reader Comment**: Additionally, the section on potential applications attracted my attention. I am a bit confused about the authors' idea to use their sampler in the Global Network of Isotopes in Precipitation (GNIP; see Section 5). Currently, it sounds as if they suggest replacing the current cumulative collectors with their automatic sampler. As far as I know, the main aim of GNIP is to collect integral samples, i.e., samples that represent the entire precipitation occurring during the collection period (usually a month). The samples are then routinely analyzed for $\delta^{18}O$, $\delta^2H$, and partly $^3H$. I am not sure how this could be achieved with the model described in the manuscript. In the current setup, one "collected sample represents the water under investigation at a given instant (integrated over 22 seconds)" (page 4, line 15-16). Maybe the authors could provide more details on the potential deployment as part of GNIP.

**Authors' response**: In our manuscript, we do not suggest replacing the cumulative collectors of the GNIP by GUARD autosamplers. We rather imply that "the application of GUARD samplers would be a cost-effective solution to supplement GNIP and/or GNIR stations" (page 10, line 16). Only in the case of "new stations too remote for regular manual sample collection" (page 10, line 17), we suggest that the GUARD autosampler "might even facilitate the installation" (page 10, lines 16-17), of additional GNIP/GNIR stations.

The predominantly monthly rainwater sampling interval applied at GNIP/GNIR stations offers the advantage of compatibility of the isotopic data from different stations. Therefore, we do not advocate ceasing this kind of operation. However, we stated that with the GUARD autosampler or similar autosamplers capable of high-frequency sampling, "much shorter sampling intervals would become possible which would enable researchers to investigate shorter-term variability in precipitation isotope systematics to improve our understanding of the underlying processes" (page 10, line 11-13). Investigating processes acting on short time-scales requires high-frequency sampling of (rain)water. To achieve this in hydrology/meteorology sampling frequency has to be at least high enough to resolve different precipitation events ("event-based" sampling). For instance, only by using such event-based data Celle-Jeanton et al. (2001) were able to demonstrate characteristic differences in the isotopic composition of rainwater in the Mediterranean coastal region of France the authors attributed to different types of synoptic weather systems. As the synoptic weather situation can change rather quickly, monthly rainwater isotope data would have most likely been of insufficient temporal resolution to identify this relationship between isotope composition and

synoptics. Interpretations on monthly rainwater isotope compositions alone can even be misleading as demonstrated by a case study conducted by Treble et al. (2005) : While monthly rainwater isotope compositions on Tasmania suggested a control by temperature (positive correlation), a 6-yr-long daily record revealed a strong amount effect as the actual mechanism controlling the isotopic composition of individual rainfall events.

**Reader Comment**: Would they still use a peristaltic pump or would the rainwater flow into the vials by gravity?

**Authors' response**: Yes, we would still use a peristaltic pump because pumping (by under- or overpressure) is necessary to inject the sample liquid into the septa-sealed sample vials as these are airtight and need to be filled through a double-canula with a small diameter for the double-canula to be able to pierce the septa. Sample injection by gravity alone is not possible due to the flow resistance exerted by the small-diameter canula.

**Reader Comment**: Would they use the same vial number (48) and size (12 mL)?

**Authors' response**: The maximum benefit from the application of the GUARD autosampler is achieved if it is equipped with the maximum number of sample vials, i.e. 160. The sample volume of 12 mL, however, is sufficient for most analyses, including isotope ratio mass spectrometry (IR-MS) and inductively coupled plasma mass spectrometry (ICP-MS).

**Reader Comment**: How would they approach programming collection intervals, without knowing when it will rain?

**Authors' response**: The temporally discontinuous nature of rainfall poses a fundamental challenge to automatic rainwater sampling. In general, in order to prevent the pump from running dry and to avoid insufficient sample volumes during sample collection, rainwater needs to be pre-collected in a suitable container. In our case studies in karst caves we applied a specifically designed pre-collection container ("pre-collector") with an internal volume of exactly 12 mL. During dripwater pre-collection a 3D-printed floating body (volume considered) inside the pre-collector would rise until it seals the pre-collector once it is completely filled with dripwater. Any dripwater in excess of 12 mL spills over through a small hole at the top of the pre-collector (Fig. 1).

[Figure]

**Fig. 1: Pre-collector used during the case studies.**

**Changes to the manuscript**: Add Fig. 1 in the authors' response to the Supplementaries and insert a brief description of its purpose and design similar to above.

**Authors' response (continued)**: One issue is that collection of rainwater needs to be initiated automatically as soon as a sufficient sample volume is available, or later, but not earlier. To ascertain that a sufficient sample volume is indeed available a detector is needed that ends hibernation and triggers sample collection. This could be achieved by implementing a photo sensor or some other kind of detector. As such a detector was not required for our case studies in karst caves but is needed for the automatic sampling of rainwater we suggest to highlight that the GUARD autosampler, at its current setup, is not suited for rainwater sampling as also proposed by Mr. Stefan Terzer-Wassmuth (reader #2).

**Changes to the manuscript**: Insert on page 10, line 17: "Due to the temporally discontinuous nature of rainfall automatic rainwater sampling requires 1) sample pre-collection for temporary storage of rainwater until a sufficient sample volume is available while minimising or even preventing evaporation and 2) a detector such as a photo sensor to end hibernation and trigger sample collection once a sufficient sample volume has been provided by rainfall. For the case studies in karst caves presented in this paper we applied a specifically designed pre-collection container ("pre-collector") with an internal volume of exactly 12 mL (Supplementary C). During dripwater pre-collection a 3D-printed floating body inside the pre-collector would rise until it seals the pre-collector once it is completely filled with dripwater. Any dripwater in excess of 12 mL spills over through a small hole at the top of the pre-collector. It is important to note that, at its current setup, the GUARD autosampler does not comprise a sample volume detector and is therefore not suited for rainwater sampling. As automatic rainwater sampling would be beneficial in numerous applications, such a detector certainly represents a useful future extension to the current GUARD system."

**Reader Comment**: Could their sampler also be used at GNIP sites exhibiting harsh conditions (i.e., a warm and arid climate)?

**Authors' response:** Yes, due to ruggedized design and the airtight sample vials we do not see a reason why the GUARD sampler should not be applicable in warm and/or arid climates, especially for sampling continuously provided media that does not require sample pre-collection. For sampling discontinuous media such as rainwater, the pre-collector should be installed inside the casing to minimise evaporation.

**Reader Comment**:

Alternatively, the authors could phrase their idea more carefully, for example by suggesting the addition of their device to the cumulative collectors at GNIP stations (instead of replacing them).

I hope these minor comments are helpful and perhaps contribute to further improvement of the manuscript, which is already a good contribution in presenting a useful automatic sampler.

**Authors' response**: To further clarify that we propose adding GUARD (or similar) samplers to the GNIP and especially the GNIR, rather than replacing the collectors currently in operation, we suggest the following changes to the manuscript as also suggested in the authors' response to reader #2 (Mr. Terzer-Wassmuth):

**Changes to the manuscript**: Replace the sentences on page 10, lines 9-14, "As mentioned in the Introduction, for this purpose the GNIP supplies researchers with isotope data generated from (mostly) monthly composite samples of rainwater collected at the ~ 1,000 GNIP stations worldwide.

If these stations were supplemented with GUARD autosamplers, much shorter sampling intervals would become possible which would enable researchers to investigate shorter-term variability in precipitation isotope systematics to improve our understanding of the underlying processes. To achieve this, sampling frequency needs to be at least high enough to resolve different precipitation events ("event-based" sampling). For instance, only by using such event-based data Celle-Jeanton et al. (2001) were able to demonstrate characteristic differences in the isotopic composition of rainwater in the Mediterranean coastal region of France the authors attributed to different types of synoptic weather systems. As the synoptic weather situation can change rather quickly, monthly rainwater isotope data would have most likely been of insufficient temporal resolution to identify this relationship between isotope composition and synoptics. Naturally, the increased number of samples generated by high-frequency sampling needs to be considered.

In addition, paraffin oil would not be required to prevent evaporation and increased maintenance of CRDS instruments could be avoided. The GUARD autosampler could also be applied at the ~ 750 stations of the Global Network for Isotopes in Precipitation (GNIR), also coordinated by the IAEA. Especially in very remote areas, the application of GUARD samplers would be a cost-effective solution to supplement GNIP and/or GNIR stations and it might even facilitate the installation of new stations too remote for regular manual sample collection."

**References**

Celle-Jeanton, H., Travi, Y., Blavoux, B., 2001. Isotopic typology of the precipitation in the Western Mediterranean region at three different time scales. Geophysical Research Letters 28, 1215–1218.

Coplen, T. B., Neiman, P. J., White, A. B., Landwehr, J. M., Ralph, M.; Dettinger, M. D. (2008) Extreme changes in stable hydrogen isotopes and precipitation characteristics in a landfalling Pacific storm, Geophys. Res. Lett., 35, L21808.

Treble, P., Budd, W.F., Hope, P.K., Rustomji, P.K., 2005b. Synoptic-scale climate patterns associated with rainfall delta O-18 in southern Australia. Journal of Hydrology 302, 270–282.

---

## Author Comment (AC4) · 8 Apr 2018

**Authors' response to short comment 2**

**Reader Comment**:

Hartmann et al. present a novel water autosampler which indeed excels over similar devices in the number of samples to be collected without supervision and also in its evaporation-protective properties (indeed most products seem to focus on dissolved constituents or their radioactivity and less on the water itself being the carrier medium). Their particular efforts to design a relatively small device in a ruggedized casing look promising for unsupervised sample collection even in remote and/or poorly accessible locations.

Key technical aspects have already been assessed by reviewer 1 and 2 (carryover effects of the peristaltic pump, eventual vulnerability towards evaporation within the tubing prior to injecting into the vial, storage effects in the Exetainer vial [note that the vendor specifications of the OA-ICO spectrometer state a typical drift of up to 0.2 per mil d18O; hence almost all tested and retested samples fall within instrument specifications]).

In line with reviewer 2 (and recognizing a number of parts in Fig. 2 from various online 'makershops)' we encourage the authors to make their work accessible to the broader scientific community in a reproducible manner, recognizing a number of additional applications after small modifications.

Notwithstanding the above, we'd like to make three remarks for the authors' kind consideration:

(1) The necessity to establish a local precipitation isotopic baseline (pg. 2 line 23-31) is undisputed and we appreciate that the Global Network of Isotopes in Precipitation (GNIP) is listed as the key resource. However, the authors present some vague assumptions regarding GNIP:

    a. not all these samples are collected manually, including some of them being totalized in active or passive devices (there are a number of devices compliant with the GNIP sampling guidelines).

    b. GNIP is *coordinated* by the IAEA, while sampling efforts are undertaken through dedicated partner institutions in IAEA or WMO Member States.

    c. To properly cite the GNIP database, pls. see http://www-naweb.iaea.org/napc/ih/IHS_resources_gnip.html (scroll down to 'obtaining and citing GNIP data'; note that Bowen and Wilkinson 2002 refer to a derivative isoscape product).

    d. Whilst the GNIP database includes sections for data sampled at other temporal resolutions than monthly, this remains the default sampling frequency to assure the worldwide compatibility of GNIP data from different sources. In settings where no permanent staffing is available at GNIP stations, a number of totalizers have been tested to compensate for this deficiency (see the GNIP manual http://www-naweb.iaea.org/napc/ih/documents/other/gnip_manual_v2.02_en_hq.pdf or Terzer et al. 2016)

**Authors' response**: We agree with all of the above statements and suggest the following changes to the manuscript:

**Changes to the manuscript:**

Replace the sentence on page 2, lines 24-26, with "The majority of such studies rely on rainwater samples (mostly) collected manually at stations of the Global Network of Isotopes in Precipitation (GNIP; IAEA/WMO, 1994) coordinated by the International Atomic Energy Agency (IAEA) with the sampling performed by dedicated partner institutions in member states of the IAEA or the World Meteorological Organisation (WMO).

Replace the sentence on page 2, lines 26-28, with "At these stations, rainwater is generally sampled at monthly resolution to ensure worldwide compatibility of GNIP data from different sources. While most of these samples are collected manually, a number of active or passive totalizers compliant

with the GNIP sampling guidelines (Terzer et al., 2016) are in operation at GNIP stations without permanent staffing. Manual sampling at higher temporal resolution, such as rainfall event-based sampling, is practically impossible as this would require round-the-clock stand-by duty."

**Reader Comment**: As a side note, paraffin wax is never used for sealing water samples during passive totalization (we assume the authors referred to paraffin oil) and its deleterious effects on laser spectrometry are subject to debate (e.g. Wassenaar et al. 2018 found that the laser spectrometry data are more vulnerable to VOC contamination; however increased spectrometer maintenance is of relevance).

We recommend to the authors to shorten the corresponding paragraph to highlighting the importance of establishing an isotope baseline for meteoric waters, to mentioning the spatial and temporal discontinuity in the GNIP database as a problem statement, and to acknowledging that sealants other than paraffin are advisable for the ease of handling and reduced need for spectrometer maintenance.

**Authors' response**: We agree with the above statement and suggest the following changes to the manuscript:

**Changes to the manuscript:** Replace the sentence on page 2, lines 28-31, with "Furthermore, sample alteration due to evaporation is commonly prevented by sealing the water samples' surface with paraffin oil despite it causing an increased need for maintenance of the standard instrument for water isotope analysis, i.e. Cavity Ring-Down Spectroscopy (CRDS).

Establishing an isotope baseline for meteoric waters is crucial for research in hydrology, meteorology and other scientific fields. While remarkable progress has been made thanks to GNIP data, the network is still spatially and temporally discontinuous, among other reasons due to the practical constraints on rainwater sampling in remote areas. Automated rainwater sampling could help solve this issue and increased maintenance of spectrometers could be avoided by applying gastight sample vials."

**Reader Comment**: (2) In line with the above, we find the authors' comments regarding potential applications at GNIP stations pretty presumptuous (pg. 10 line 8 ff.):

a. the 1,000 sites mentioned are not temporally continuous, and the network is *coordinated* by IAEA, however the sampling is carried out by partner institutions in IAEA or WMO Member States, and the choice of equipment is subject to their respective organizational context as long as it is compliant with the GNIP sampling guidelines.

b. as indicated, the presently active GNIP stations employ an array of sampling methods (yet compliant with the protocol), not all of which rely on daily retrieval of the rainwater and not all totalizing stations employing paraffin oil as a sealant.

We agree however that a coordinated effort of sampling at higher frequency may be beneficial to a number of especially meteorological and climatological assessments; however the resulting analytical effort needs to be kept in mind.

**Authors' response**: We agree with the above statements and suggest the following changes to the manuscript:

**Changes to the manuscript:** Replace the sentences on page 10, lines 9-14, "As mentioned in the Introduction, for this purpose the GNIP supplies researchers with isotope data generated from

(mostly) monthly composite samples of rainwater collected at the ~ 1,000 GNIP stations worldwide. If these stations were supplemented with GUARD autosamplers, much shorter sampling intervals would become possible which would enable researchers to investigate shorter-term variability in precipitation isotope systematics to improve our understanding of the underlying processes. To achieve this, sampling frequency needs to be at least high enough to resolve different precipitation events ("event-based" sampling). For instance, only by using such event-based data Celle-Jeanton et al. (2001) were able to demonstrate characteristic differences in the isotopic composition of rainwater in the Mediterranean coastal region of France the authors attributed to different types of synoptic weather systems. As the synoptic weather situation can change rather quickly, monthly rainwater isotope data would have most likely been of insufficient temporal resolution to identify this relationship between isotope composition and synoptics. Naturally, the increased number of samples generated by high-frequency sampling needs to be considered.

In addition, paraffin oil would not be required to prevent evaporation and increased maintenance of CRDS instruments could be avoided. The GUARD autosampler could also be applied at the ~ 750 stations of the Global Network for Isotopes in Precipitation (GNIR), also coordinated by the IAEA. Especially in very remote areas, the application of GUARD samplers would be a cost-effective solution to supplement GNIP and/or GNIR stations and it might even facilitate the installation of new stations too remote for regular manual sample collection."

**Reader Comment**: (3) On the technical side, we see that the sampler in its present form may certainly be applied to sample continuously flowing media (drip waters, groundwaters, surface waters, leachates etc.), however given the discontinuous nature of precipitation in both timing and intensity, and the resulting need to integrate into discrete, timed samples (regardless of whether samples are taken on a sub-hourly, daily, or monthly basis), the device in its present form is not undisputedly suited as a precipitation sampler since it lacks (a) a precipitation trigger to end hibernation mode, and (b) appropriate means to totalize precipitation, safe from evaporation, over the sampling interval prior to dispensing into the vial.

Based on (2) and (3) we suggest to the authors to shorten the paragraph on potential applications, and we strongly advise to state that the device in the form presented is not capable of collecting discontinuous media such as precipitation (but this could be added as an outlook).

**Authors' response**: As with a similar comment by reader #1 (Mr. Michelsen) we agree with the above statement and suggest the following changes to the manuscript already suggested in the authors' response to the comments of reader #1:

The temporally discontinuous nature of rainfall poses a fundamental challenge to automatic rainwater sampling. In general, in order to prevent the pump from running dry and to avoid insufficient sample volumes during sample collection, rainwater needs to be pre-collected in a suitable container. In our case studies in karst caves we applied a specifically designed pre-collection container ("pre-collector") with an internal volume of exactly 12 mL. During dripwater pre-collection a 3D-printed floating body (volume considered) inside the pre-collector would rise until it seals the pre-collector once it is completely filled with dripwater. Any dripwater in excess of 12 mL spills over through a small hole at the top of the pre-collector (Fig. 1).

[Figure]

**Fig. 1: Pre-collector used during the case studies.**

**Changes to the manuscript**: Add Fig. 1 in the authors' response to the Supplementaries and insert a brief description of its purpose and design similar to above.

**Authors' response (continued):** One issue is that collection of rainwater needs to be initiated automatically as soon as a sufficient sample volume is available, or later, but not earlier. To ascertain that a sufficient sample volume is indeed available a detector is needed that ends hibernation and triggers sample collection. This could be achieved by implementing a photo sensor or some other kind of detector. As such a detector was not required for our case studies in karst caves but is needed for the automatic sampling of rainwater we suggest to highlight that the GUARD autosampler, at its current setup, is not suited for rainwater sampling as proposed by Mr. Stefan Terzer-Wassmuth (reader #2).

**Changes to the manuscript:** Insert on page 10, line 17: "Due to the temporally discontinuous nature of rainfall automatic rainwater sampling requires 1) sample pre-collection for temporary storage of rainwater until a sufficient sample volume is available while minimising or even preventing evaporation and 2) a detector such as a photo sensor to end hibernation and trigger sample collection once a sufficient sample volume has been provided by rainfall. For the case studies in karst caves presented in this paper we applied a specifically designed pre-collection container ("pre-collector") with an internal volume of exactly 12 mL (Supplementary C). During dripwater pre-collection a 3D-printed floating body inside the pre-collector would rise until it seals the pre-collector once it is completely filled with dripwater. Any dripwater in excess of 12 mL spills over through a small hole at the top of the pre-collector. It is important to note that, at its current setup, the GUARD autosampler does not comprise a sample volume detector and is therefore not suited for rainwater sampling. As automatic rainwater sampling would be beneficial in numerous applications, such a detector certainly represents a useful future extension to the current GUARD system."

**Reader Comment**: To conclude, we congratulate the authors on their achievements with the development of an autosampler for continuously flowing media, and we look forward to see the concept expanded into discontinuous media as well.

**References**

Celle-Jeanton, H., Travi, Y., Blavoux, B., 2001. Isotopic typology of the precipitation in the Western Mediterranean region at three different time scales. Geophysical Research Letters 28, 1215–1218.

IAEA/WMO, 1994, Global Network for Isotopes in Precipitation (GNIP) Database. IGBP PAGES/World Data Center-A for Paleoclimatology Data Contribution Series # 94-005.  NOAA/NGDC Paleoclimatology Program, Boulder CO, USA.

Terzer, S., et al., 2016. An assessment of the isotopic (2H/18O) integrity of water samples collected and stored by unattended precipitation totalizers. Geophysical Research Abstracts, 18, EGU2016-15992.

Wassenaar, L.I. et al., 2018. Seeking excellence: An evaluation of 235 international laboratories conducting water isotope analyses by isotope-ratio and laser-absorption spectrometry. Rapid Communications in Mass Spectrometry 32, 393-406

---

## Author Response (AR2)

**Authors' response to final editor decision**

**Editor:**

Dear authors,

thanks for the detailed revisions and the revised manuscript.

**Editor comment 1:** I agree with most of them, however, I would propose to expand Table 2 of the manuscript to include the other rain collectors (OPEnSampler, Lisa Liquidsampler, Sampler by Coolen) available in addition to the ISCO 3700. This would certainly help the reader to see the novelty of the GUASRD sampler and to make an informed decision about which sampler could be used for their study.

**Authors' response**: We have expanded Table 2 as proposed by the Editor. Unfortunately, some of the information to be indicated in the Table are not publicly available. Retrieving the missing information, if possible at all, by contacting the developers of the other liquid autosamplers would have prevented us from keeping the upload deadline. If the missing information is considered crucial, the deadline will have to be postponed. For simplicity, we have uploaded the revised Table 2 as a separate PDF file.

**Editor comment 2:** I also had a detailed look at the proposed information of open science and reproducibility. I am not completely happy with the proposed tables, since a step-by-step guide is missing, which could be made available in GitHub for example - so there is no need to include this into the supplementary materials.

**Authors' response**: Taking into account our limited capacities, we have created assembly instructions to the best of our abilities so that other members of the scientific community and interested members of the public may be enabled to build GUARD autosamplers of their own. The assembly instructions together with all other necessary documents (CAD file and Code file) may be uploaded to a file sharing platform such as GitHub as suggested by the Editor. For the review process we have uploaded the assembly instructions as well as the Code as one combined PDF file. The CAD file can be sent separately by E-mail, if necessary.

**Editor:**

Please provide a detailed response to each reviewer comment and also provide a revised version of the manuscript with the changes marked.

Best regards

Markus Weiler